# Efficiently Learning Probabilistic Logical Models by Cheaply Ranking Mined Rules

## Abstract

Probabilistic logical models are a core component of neurosymbolic AI and are important models in their own right for tasks that require high explainability. Unlike neural networks, logical models are often handcrafted using domain expertise, making their development costly and prone to errors. While there are algorithms that learn logical models from data, they are generally prohibitively expensive, limiting their applicability in real-world settings. In this work, we introduce precision and recall for logical rules and define their composition as rule utility – a cost-effective measure to evaluate the predictive power of logical models. Further, we introduce SPECTRUM, a scalable framework for learning logical models from relational data. Its scalability derives from a linear-time algorithm that mines recurrent structures in the data along with a second algorithm that, using the cheap utility measure, efficiently ranks rules built from these structures. Moreover, we derive theoretical guarantees on the utility of the learnt logical model. As a result, we demonstrate across various tasks that SPECTRUM scales to larger datasets, often learning more accurate logical models orders of magnitude faster than previous methods without requiring specialised GPU hardware.

## 1 Introduction

**Motivation.** *Neurosymbolic AI* combines neural networks with (probabilistic) logical models, to harness the strengths of both approaches (d'Avila Garcez et al., 2019; d'Avila Garcez et al., 2022). Neurosymbolic frameworks outperform neural networks in several areas (Manhaeve et al., 2018; Gu et al., 2019; Mao et al., 2019), particularly in interpretability (Mao et al., 2019) and in reducing the need for data (Feldstein et al., 2023a). Unlike neural networks, which are trained from data, logical models are typically handcrafted. Thus, developing logical models requires domain expertise, in both the data and the inference task. This process is costly and prone to errors. As a result, there has been increased attention on learning logical models from data – a task known as *structure learning*.

**Limitations of state-of-the-art.** Examples of probabilistic logical models include *Markov logic networks* (MLNs) (Richardson & Domingos, 2006), *probabilistic soft logic* (PSL) (Bach et al., 2017) and *probabilistic logical programs* (PLPs) (Poole, 1993; Sato, 1995; De Raedt et al., 2007). Numerous structure learning techniques for MLNs (Mihalkova & Mooney, 2007; Kok & Domingos, 2010; Khot et al., 2015; Feldstein et al., 2023b) and especially PLPs (Quinlan, 1990; Muggleton, 1995; Evans & Grefenstette, 2018; Schüller & Benz, 2018; Qu et al., 2021b; Cheng et al., 2023) have been proposed. However, an overarching limitation of structure learning remains the limited scalability to large datasets. The underlying difficulty is the exponential nature of the problem with respect to the length of possible rules and the number of relations in the data. State-of-the-art structure learning algorithms aim to reduce the complexity of the problem by splitting the task into two steps: 1) *Pattern mining* - which identifies frequently occurring substructures in the data. 2) *Optimisation* - an iterative process during which the best logical formulae are chosen from a set of candidates identified within the patterns. Any structure learning algorithm must make approximations to tackle scalability. Existing methods proceed by finding *approximate* patterns but perform *exact* inference.

**Contributions.** Our key idea is to flip the aforementioned paradigm: find *exact* patterns but perform *approximate* ranking of candidate formulae to create the final logical model. By eliminating

a combinatorial explosion of inference steps, this strategy effectively improves scalability by orders of magnitude. Specifically, we present three contributions to tackle scalability:

1. *Cheap utility measure*: We introduce *utility*, a criteria for measuring the predictive power of individual rules and logical theories. The utility relates to the degree to which a rule or logical model is satisfied in the data (*precision*) as well as how often the rules predict data points (*recall*). This measure can be computed cheaply, without requiring exact inference.

2. *Linear-time pattern mining*: We present a linear-time algorithm (in the size of the dataset) that finds the number of occurrences of different patterns in the data. The algorithm is approximate, yet we provide theoretical guarantees on the computational cost required for a certain error bound on the utility estimates. These guarantees are dataset-independent.

3. *Quadratic-time optimisation*: We present a quadratic-time greedy optimisation algorithm (in the number of rules of the final model). The algorithm finds the best rules and sorts them by their utility. Since the utility measure is cheap to evaluate, compared to prior art this optimisation step is no longer a bottleneck for scalability.

**Empirical results.**   In addition to the theoretical contributions, we present SPECTRUM, a parallelised C++ implementation of our structure learning framework. We show on various relational datasets that SPECTRUM improves scalability by orders of magnitude, consistently reducing runtimes to $< 1\%$ compared to the previous state-of-the-art. Also, despite minor restrictions on mined logical formulae, we find logical models that in most cases improve accuracy compared to prior art.

**Restrictions.**   SPECTRUM is restricted to datasets with unary and binary relations, however, many real-world datasets fit within this restriction. In addition, our utility measure is only well defined for Datalog theories (Abiteboul et al., 1995), a language used extensively in data management (Barceló & Pichler, 2012; Moustafa et al., 2016) and neurosymbolic learning (Huang et al., 2021).

## 2   PRELIMINARIES

**First-order logic.**   In first-order logic, *constants* represent objects in the domain (e.g. `alice`, `bob`). *Variables* range over the objects (e.g. $X$, $Y$, $Z$). A *term* is a constant or a variable. A predicate P represents a relation between objects (e.g. FRIENDS). The *arity* of P is the number of related objects. An *atom* has the form $\text{P}(t_1, \ldots, t_n)$, where P is an $n$-ary predicate, and $t_1, \ldots, t_n$ are terms. A *fact* is an atom for which each term $t_i$ is a constant (e.g. FRIENDS(`alice`,`bob`)). A *relational database* $\mathcal{D}$ is a set of facts. A Datalog rule $\rho$, or simply *rule*, is a first-order logic formula of the form $\forall \mathbf{X}. \bigwedge_{i=1}^{n} \text{P}_i(\mathbf{X}_i) \to \text{P}(\mathbf{Y})$, where $\bigwedge_{i=1}^{n} \text{P}_i(\mathbf{X}_i)$ is a *conjunction* of atoms, $\to$ denotes logical *implication*, $\mathbf{X}$, $\mathbf{X}_i$, and $\mathbf{Y}$ are tuples of variables, $\mathbf{Y} \subseteq \bigcup_{i=1}^{n} \mathbf{X}_i$, where $\mathbf{X}_i \subseteq \mathbf{X}$. Quantifiers are commonly omitted. The left-hand and the right-hand side of a rule are its *body* and *head*, respectively, and are denoted by $\text{body}(\rho)$ and $\text{head}(\rho)$. The *length* of a conjunction is the number of its conjuncts. The *length* of a rule $L(\rho)$ is the length of its body plus the length of its head. A *grounding* of an atom is the atom that results after replacing each occurrence of a variable with a constant. *Groundings* of conjunctions and rules are defined analogously. A *theory* $\boldsymbol{\rho}$ is a set of rules. In probabilistic logic models, the rules are associated with a weight, where the weight represents the likelihood of the rule being satisfied (Richardson & Domingos, 2006; Bach et al., 2017).

**Hypergraphs.**   A *hypergraph* $\mathcal{G}$ is a pair of the form $(V, E)$, where $V$ is a set of nodes and $E$ is a set of edges with each element of $E$ being a set of nodes $\{v_1, \ldots, v_n\}$ from $V$. A hypergraph $\mathcal{G}$ is *labelled* if each edge $e$ in $\mathcal{G}$ is labelled with a categorical value denoted by $\text{label}(e)$. A *path* in a hypergraph is an alternating sequence of nodes and edges, $(v_1, e_1, \ldots, e_l, v_{l+1})$, where each edge $e_i$ contains $v_i$ and $v_{i+1}$. The *length* of a path is the number of edges in the path. A hypergraph is *connected* if there exists a path between any two nodes. The *distance* between two nodes is the length of the shortest path that connects them. A relational database $\mathcal{D}$ can be represented by a hypergraph $\overline{\mathcal{G}}_{\mathcal{D}} = (V, E)$ where, for each constant $\text{c}_i$ occurring in $\mathcal{D}$, $V$ includes a node, and, for each fact $\text{P}(\text{c}_1, \ldots, \text{c}_k)$ in $\mathcal{D}$, $E$ includes an edge $e = \{\text{c}_1, \ldots, \text{c}_k\}$ with label P. Two graphs $\mathcal{G}_1$ and $\mathcal{G}_2$ are *isomorphic* if there exists a one-to-one mapping $I$ from the nodes of $\mathcal{G}_1$ to the nodes of $\mathcal{G}_2$ so that for each edge $e = \{v_1, \ldots, v_k\}$ in $\mathcal{G}_1$, $e' = \{I(v_1), \ldots, I(v_k)\}$ is an edge in $\mathcal{G}_2$, and vice versa. For brevity, from now on, we refer to hypergraphs simply as graphs.

## 3 PATTERNS

We introduce the concept of patterns - commonly recurring substructures within a database graph. From now on, we assume that $\mathcal{D}$ is fixed and clear from context. As stated in the introduction, we restrict our discussions to graphs with only unary and binary edges. We use $\mathsf{u}_\mathcal{G}(v)$ and $\mathsf{b}_\mathcal{G}(v)$ to denote the set of unary and binary edges in the graph $\mathcal{G}$ that are incident to $v$. Further, we use $\boldsymbol{\alpha}$ to denote the set of atoms that have a grounding in $\mathcal{D}$ and $\alpha \in \boldsymbol{\alpha}$ to denote a particular atom. We use $\overline{\boldsymbol{\alpha}}$ to denote the set of all possible groundings of $\alpha$ in $\mathcal{D}$ and $\overline{\alpha} \in \overline{\boldsymbol{\alpha}}$ to denote a particular grounding.

Similarly to how we can view relational databases as graphs, we can also view conjunctions of atoms as graphs. For a conjunction of atoms $\varphi := \bigwedge_{i=1}^{n} \mathsf{P}_i(\mathbf{t}_i)$, the *pattern* of $\varphi$, denoted as $\mathcal{G}_\varphi = (V, E)$, is the graph where, for each term $t_i$ occurring in $\varphi$, $V$ includes a node $t_i$, and, for each atom $\mathsf{P}(t_1, \ldots, t_n)$ occurring in $\varphi$, $E$ includes an edge $\{t_1, \ldots, t_n\}$ with label $\mathsf{P}$. The *length* of a pattern $\mathcal{G}_\varphi$ is the number of atoms in $\varphi$. Given a rule $\rho$, the patterns corresponding to its head and body are denoted by $\mathcal{G}_{\mathsf{head}(\rho)}$ and $\mathcal{G}_{\mathsf{body}(\rho)}$, respectively. We call $\mathcal{G}_{\mathsf{body}(\rho) \wedge \mathsf{head}(\rho)}$ the *rule pattern* of $\rho$. Rule $\rho$ is *connected* if $\mathcal{G}_{\mathsf{body}(\rho) \wedge \mathsf{head}(\rho)}$ is connected; it is *body-connected* if $\mathcal{G}_{\mathsf{body}(\rho)}$ is connected.

A *ground pattern* of a conjunction $\varphi$ is the graph corresponding to a grounding of $\varphi$ that is satisfied in $\mathcal{D}$. We denote by $\overline{\boldsymbol{\mathcal{G}}}_{\mathsf{body}(\rho) \wedge \mathsf{head}(\rho)}$ the set of all ground patterns of $\mathsf{body}(\rho) \wedge \mathsf{head}(\rho)$ in $\mathcal{D}$. For a fact $\overline{\alpha}$ that is a grounding of $\alpha = \mathsf{head}(\rho)$, we use $\overline{\boldsymbol{\mathcal{G}}}_{\mathsf{body}(\rho) \wedge \mathsf{head}(\rho)}^{\mathsf{head}(\rho) = \overline{\alpha}}$ to denote the subset of $\overline{\boldsymbol{\mathcal{G}}}_{\mathsf{body}(\rho) \wedge \mathsf{head}(\rho)}$ which contains only groundings of patterns of the form $\mathcal{G}_{\mathsf{body}(\rho) \wedge \overline{\alpha}}$.

## 4 RULE UTILITY

In this section, we introduce a measure, that we call *utility*, for assessing the "usefulness" of a theory without requiring inference. The utility itself depends on various criteria, which we present below. The following definitions hold for connected rules that are also body connected.

**Definition 1** (Precision). *The precision of rule $\rho$ is defined as* $\mathsf{P}(\rho) := \frac{|\overline{\boldsymbol{\mathcal{G}}}_{\mathsf{body}(\rho) \wedge \mathsf{head}(\rho)}|}{|\overline{\boldsymbol{\mathcal{G}}}_{\mathsf{body}(\rho)}|}$.

Intuitively, $\mathsf{P}(\rho)$ is thus the fraction of times that the head and body of a rule are both true in the data when the body is true in the data[1]. If one considers cases where the body and head are both true as true positives (TP, i.e. the rule is satisfied), and cases where the body is true and the head is false as false positives (FP, i.e. the rule is not satisfied), then the definition of precision is analogous to the definition of precision in classification tasks, i.e. $\mathsf{P}(\rho) = TP/(TP + FP)$. Useful rules should make claims that are often true, and thus have high precision.

One issue with precision, as defined above, is that it underestimates how often a rule is satisfied if there are symmetries in the rule. We fix this issue, which we illustrate graphically in Appendix B, by multiplying the precision by a symmetry factor:

**Definition 2** (Symmetry factor). *The symmetry factor of rule $\rho$, denoted by $\mathsf{S}(\rho)$, is defined as the number of subgraphs in $\mathcal{G}_{\mathsf{body}(\rho) \wedge \mathsf{head}(\rho)}$ that are isomorphic to $\mathcal{G}_{\mathsf{body}(\rho)}$.*

The second issue with precision, as defined above, is that, if the facts in $\mathcal{D}$ are unbalanced (i.e. facts of different predicates occur with different frequencies), then certain rules can still have high precision even if the facts are uncorrelated. We illustrate this issue with an example in Appendix B. We fix this issue by dividing the precision by a Bayesian prior:

**Definition 3** (Bayesian Prior). *The Bayesian prior of rule $\rho$ is defined as* $\mathsf{B}(\rho) := \frac{|\overline{\boldsymbol{\mathcal{G}}}_{\mathsf{head}(\rho)}|}{\sum_{\alpha \in \mathcal{A}} |\overline{\boldsymbol{\mathcal{G}}}_\alpha|}$, *where $\mathcal{A}$ is the set of all atoms constructable over all predicates in $\mathcal{D}$ of the same tuple of terms as $\mathsf{head}(\rho)$.*

The symmetry and prior-corrected precision is the product $\frac{\mathsf{P}(\rho) \cdot \mathsf{S}(\rho)}{\mathsf{B}(\rho)}$. A useful rule should have a symmetry-corrected precision, $\mathsf{P}(\rho) \cdot \mathsf{S}(\rho)$, that is better than random chance, $\mathsf{B}(\rho)$, i.e. $\frac{\mathsf{P}(\rho) \cdot \mathsf{S}(\rho)}{\mathsf{B}(\rho)} > 1$.

In addition to being precise, useful rules should account for many diverse observations in the data. Below, we introduce a metric to count how often a rule pattern recalls facts in the data.

---

[1]Definition 1 is equivalent to the definition of *precision* in Gao et al. (2024) and the definition of *confidence* in Lajus et al. (2020). However, note that both these works neglect to account for symmetry and priors.

**Definition 4** (Recall). *The recall of rule $\rho$ is defined as* $\mathsf{R}(\rho) := \sum_{\overline{\alpha} \in \overline{\boldsymbol{\alpha}}} \ln(1 + |\overline{\boldsymbol{\mathcal{G}}}_{\mathsf{body}(\rho) \wedge \mathsf{head}(\rho)}^{\mathsf{head}(\rho) = \overline{\alpha}}|),$ *where $\overline{\boldsymbol{\alpha}}$ is the set of all groundings of $\alpha := \mathsf{head}(\rho)$ in $\mathcal{D}$.*

Intuitively, $|\overline{\boldsymbol{\mathcal{G}}}_{\mathsf{body}(\rho) \wedge \mathsf{head}(\rho)}^{\mathsf{head}(\rho) = \overline{\alpha}}|$ says how many different groundings of $\rho$ entail $\overline{\alpha}$, and the logarithm reflects the diminishing returns of information when recalling the same fact. Importantly, recalling the same fact increases recall logarithmically, while recalling different facts increases recall linearly.

Longer rules are biased to have more groundings in the data than shorter rules, due to a combinatorial explosion. Therefore, an issue with recall, as defined above, is that it biases towards longer rules. Longer rules increase computational costs, forcing logical solvers to make more approximations during inference, which reduces accuracy. To address this, we introduce a complexity factor that penalises longer rules:

**Definition 5** (Complexity factor). *The complexity factor of $\rho$ of length $L(\rho)$ is defined as* $\mathsf{C}(\rho) := e^{-L(\rho)}.$

The complexity-corrected recall, $\mathsf{R}(\rho) \cdot \mathsf{C}(\rho)$, discourages using longer rules if they do not have a correspondingly larger recall, thus favouring the simplest explanation for the data (Occam's razor).

Useful rules should exhibit both high precision (corrected for symmetry and prior probabilities) and high recall (corrected for rule complexity). This leads to a natural metric for quantifying the "usefulness" of a rule:

**Definition 6** (Rule utility). *The utility of rule $\rho$ is defined as* $\mathsf{U}(\rho) := \frac{\mathsf{P}(\rho)\mathsf{S}(\rho)}{\mathsf{B}(\rho)} \cdot \mathsf{R}(\rho)\mathsf{C}(\rho).$

Finally, we extend the notion of utility to a theory $\boldsymbol{\rho}$. Different rules can recall the same fact. The recall for a set of rules should be analogous to Definition 4 but include contributions from all rules:

**Definition 7** (Complexity-corrected rule-set recall). *For a set of rules $\boldsymbol{\rho}_\alpha$ having the same head $\alpha$, the complexity-corrected rule-set recall is defined as the product $\mathsf{R}(\boldsymbol{\rho}_\alpha) \cdot \mathsf{C}(\boldsymbol{\rho}_\alpha)$, where*

$$\mathsf{R}(\boldsymbol{\rho}_\alpha) := \sum_{\overline{\alpha} \in \overline{\boldsymbol{\alpha}}} \ln \left( 1 + \sum_{\rho \in \boldsymbol{\rho}_\alpha} |\overline{\boldsymbol{\mathcal{G}}}_{\mathsf{body}(\rho) \wedge \mathsf{head}(\rho)}^{\mathsf{head}(\rho) = \overline{\alpha}}| \right) \quad and \quad \mathsf{C}(\boldsymbol{\rho}_\alpha) := \left( \prod_{\rho \in \boldsymbol{\rho}_\alpha} \mathsf{C}(\rho) \right)^{\frac{1}{|\boldsymbol{\rho}|}}.$$

Intuitively, $\sum_{\rho \in \boldsymbol{\rho}_\alpha} |\overline{\boldsymbol{\mathcal{G}}}_{\mathsf{body}(\rho) \wedge \mathsf{head}(\rho)}^{\mathsf{head}(\rho) = \overline{\alpha}}|$ counts the number of different instantiations of *all* rules in the set $\boldsymbol{\rho}_\alpha$ that entail a particular fact $\overline{\alpha}$, whereas $\mathsf{C}(\boldsymbol{\rho}_\alpha)$ is simply the geometric average of the complexity factor for all rules in set $\boldsymbol{\rho}_\alpha$. We are now ready to introduce the notion of theory utility:

**Definition 8** (Theory utility). *The utility of theory $\boldsymbol{\rho}$ is defined as* $\mathsf{U}(\boldsymbol{\rho}) := \sum_{\alpha \in \boldsymbol{\alpha}} \left( \sum_{\rho \in \boldsymbol{\rho}_\alpha} \frac{\mathsf{P}(\rho)\mathsf{S}(\rho)}{\mathsf{B}(\rho)} \right) \cdot \mathsf{R}(\boldsymbol{\rho}_\alpha)\mathsf{C}(\boldsymbol{\rho}_\alpha),$ *where* $\boldsymbol{\alpha} = \{\mathsf{head}(\rho) \,|\, \rho \in \boldsymbol{\rho}\}$ *and* $\boldsymbol{\rho}_\alpha = \{\rho \in \boldsymbol{\rho} \,|\, \mathsf{head}(\rho) = \alpha\}.$

The outer sum in Definition 8 runs over the different atoms occurring in the heads of the rules in $\boldsymbol{\rho}$, while the inner sum runs over the different rules with the same head atom. Computing rule utility requires enumerating all ground patterns of a rule, its body, and its head in the data. In the next section, we outline how we find these groundings efficiently.

## 5 PATTERN MINING

In this section, we present our technique for mining rule patterns from relational data, i.e. finding subgraphs in a relational database graph. Since finding all subgraphs is generally a hard problem with no known polynomial algorithm (Bomze et al., 1999), we adopt an approach similar to PRISM (Feldstein et al., 2023b). In particular, we present a non-exhaustive algorithm that has only linear-time complexity in the dataset size but that allows us to compute estimates of utility that are *close*, in a precise sense, to their true values.

### 5.1 ALGORITHM

The steps of our technique are outlined in Algorithm 1. The algorithm mines ground patterns by calling a recursive function (NEXTSTEP) from each node $v_0$ in $\overline{\mathcal{G}}_{\mathcal{D}}$ (lines 1-3). Intuitively, the

algorithm searches for ground patterns by rolling out paths in parallel from a starting node. The recursion stops at a user-defined *maximum depth $D$*, rolling out up to a *maximum number of paths $N$*.

---

**Algorithm 1:** MINEPATTERNS($\overline{\mathcal{G}}_{\mathcal{D}}, D, N$)

**Input:** $\overline{\mathcal{G}}_{\mathcal{D}}$ – Graph representation of $\mathcal{D}$
**Output:** $\overline{\mathcal{G}}_{\text{global}}$ – global variable storing all mined ground patterns
**Parameters:** $D$ – maximum recursion depth
                   $N$ – maximum number of paths

1 **for each** $v_0$ **in** $\overline{\mathcal{G}}_{\mathcal{D}}$ **do**
2     $\overline{\mathcal{G}}_{\text{global}} \leftarrow \overline{\mathcal{G}}_{\text{global}} \cup$ DOUBLEUNARYPATTERNS($v_0$)
3     NEXTSTEP($\overline{\mathcal{G}}_{\mathcal{D}}, v_0, D, N, d = 0, \overline{\mathcal{G}}_{\text{previous}} = \{\emptyset\}, \mathcal{E}_{\text{previous}} = \emptyset$)

4 **return** $\overline{\mathcal{G}}_{\text{global}}$

**Function** NEXTSTEP ($\overline{\mathcal{G}}_{\mathcal{D}}, v, D, n, d, \overline{\mathcal{G}}_{\text{previous}}, \mathcal{E}_{\text{previous}}$):
     /* datagraph $\overline{\mathcal{G}}_{\mathcal{D}}$, current node $v$, maximum recursion depth $D$,
        maximum number of remaining paths $n$, current recursion
        depth $d$, previously found patterns $\overline{\mathcal{G}}_{\text{previous}}$, previously
        visited edges $\mathcal{E}_{\text{previous}}$                               */
5     $\overline{\mathcal{G}}_{\text{new}} \leftarrow \emptyset$
6     **for each** $\overline{\mathcal{G}}$ **in** $\overline{\mathcal{G}}_{\text{previous}}$ **do**
7        **for each** $e$ **in** $u_{\overline{\mathcal{G}}_{\mathcal{D}}}(v)$ **do**
8          $\overline{\mathcal{G}}_{\text{new}} \leftarrow \overline{\mathcal{G}}_{\text{new}} \cup \{\overline{\mathcal{G}} \circ e\}$            // Graft unary edges of $v$
9     $\overline{\mathcal{G}}_{\text{global}} \leftarrow \overline{\mathcal{G}}_{\text{global}} \cup \overline{\mathcal{G}}_{\text{new}}$
10    **if** $d < D$ **then**
11       $\mathcal{E}' \leftarrow b_{\overline{\mathcal{G}}_{\mathcal{D}}}(v) \setminus \mathcal{E}_{\text{previous}}$
12       **if** $n < |\mathcal{E}'|$ **then**
13         $\mathcal{E}' \leftarrow$ SELECT_$n$_DIFFERENT_RANDOM_ELEMENTS($\mathcal{E}', n$)
14         $n' \leftarrow 1$
15       **else**
16         $n' \leftarrow \lceil n/|\mathcal{E}'| \rceil$
17       **for each** $e := \{v, v'\}$ **in** $\mathcal{E}'$ **do**
18         $\overline{\mathcal{G}}_{\text{final}} \leftarrow \emptyset$
19         **for each** $\overline{\mathcal{G}}$ **in** $\overline{\mathcal{G}}_{\text{new}}$ **do**
20           $\overline{\mathcal{G}}_{\text{final}} \leftarrow \overline{\mathcal{G}}_{\text{final}} \cup \{\overline{\mathcal{G}} \circ e\}$        // Graft binary edges of $v$
21         $\overline{\mathcal{G}}_{\text{global}} \leftarrow \overline{\mathcal{G}}_{\text{global}} \cup \overline{\mathcal{G}}_{\text{final}}$
22         NEXTSTEP($\overline{\mathcal{G}}_{\mathcal{D}}, v', D, n', d + 1, \overline{\mathcal{G}}_{\text{final}}, \mathcal{E}_{\text{previous}} \cup \{e\}$)

---

In each call of NEXTSTEP, the algorithm visits a node $v \in V$ of $\overline{\mathcal{G}}_{\mathcal{D}}$. At node $v$, unary relations $u_{\overline{\mathcal{G}}_{\mathcal{D}}}(v)$ are grafted onto previously found ground patterns $\overline{\mathcal{G}}_{\text{previous}}$ (lines 6-8). We use $\overline{\mathcal{G}} \circ e$ to denote the graph that results after adding edge $e$ and the nodes of $e$ to graph $\overline{\mathcal{G}}$. The resulting ground patterns are stored in $\overline{\mathcal{G}}_{\text{new}}$ (line 9). If the maximum recursion depth has not been reached (line 10), a subset of the binary edges of node $v$ is then selected (lines 11-16). The algorithm avoids mining patterns corresponding to tautologies by excluding previously visited binary edges (line 11). To keep the complexity linear, we enforce $N$ to be the maximum number of paths by setting the maximum number of selected binary edges $n$ to be $N$ divided by the number of binary edges selected at each previous stage (lines 13, 16). We graft each chosen binary edge onto the ground patterns in $\overline{\mathcal{G}}_{\text{new}}$ (lines 17-20), store the new ground patterns in $\overline{\mathcal{G}}_{\text{final}}$ (line 21), and pass $\overline{\mathcal{G}}_{\text{final}}$ on to the subsequent call (line 22). $\overline{\mathcal{G}}_{\text{final}}$ is passed to the next recursive call to continuously extend previously mined patterns. In the next recursive call, the recursion depth $d$ is increased by 1, thereby expanding the search of grounds patterns to include nodes up to a distance $d$ away from $v_0$. At each depth $d \in \{0, \ldots, D-1\}$ the algorithm grafts up to one unary and one binary onto the patterns previously discovered along that path. At depth $d = D$, the algorithm only grafts up to

one unary onto the patterns, since it terminates before grafting binaries (line 10). Thus, the patterns found by Algorithm 1 are of maximum length $2D + 1$.

As a special case, the algorithm also mines patterns that consist of two unary edges on a single node in line 2, to allow constructing rules of the form $P_1(X) \rightarrow P_2(X)$. DOUBLEUNARYPATTERNS creates all possible patterns that consist of all pairings of distinct unary edges from the set $u_{\overline{\mathcal{G}}_{\mathcal{D}}}(v_0)$.

**Remark 1.** *For simplicity, in Algorithm 1, we presented $\overline{\mathcal{G}}_{\text{global}}$ as a set of ground patterns. However, in a later stage of our structure-learning algorithm, we will also need knowledge of the corresponding non-ground patterns to compute rule utility (Section 4). In our implementation, $\overline{\mathcal{G}}_{\text{global}}$ is thus in fact a map from patterns to every corresponding ground pattern that was found in $\overline{\mathcal{G}}_{\mathcal{D}}$. This map is generated on the fly, where everytime a new ground pattern is mined, we obtain its corresponding pattern by variabilising its constants (up to isomorphism) and then adding it to the map.*

## 5.2 THEORETICAL PROPERTIES

This section presents the complexity of Algorithm 1 and provide completeness guarantees, as well as guarantees on the uncertainty of the mined patterns. Proofs of all theorems are given in Appendix A.

**Theorem 1** (Completeness). *Let $v$ and $v'$ be two nodes in $\overline{\mathcal{G}}_{\mathcal{D}}$ that are distance $l$ apart, for some $l \geq 0$. We say that $v'$ is $N$-close to $v$ if, for each path $(v_{i_0}, e_{i_0}, \ldots, e_{i_{l-1}}, v_{i_l})$ of length $l$ between $v$ and $v'$ in $\overline{\mathcal{G}}_{\mathcal{D}}$, where $v_{i_0} = v$ and $v_{i_l} = v'$, the following holds: $|b_{\overline{\mathcal{G}}_{\mathcal{D}}}(v_{i_0})| \prod_{j=1}^{l-1} (|b_{\overline{\mathcal{G}}_{\mathcal{D}}}(v_{i_j})| - 1) \leq N$.*

*Then, for each $N \geq 0$, each $D \geq 0$, and each $v \in \overline{\mathcal{G}}_{\mathcal{D}}$, Algorithm 1 mines* all *ground patterns involving $v$ and nodes that are $N$-close to $v$ and a distance $\leq D$ from $v$; all remaining ground patterns involving $v$ and nodes within distance $D$ are found with a probability larger than when running $N$ random walks from $v$.*

When mining patterns, Algorithm 1 runs at most $N$ paths from $|V|$ nodes up to a maximum recursion depth $D$. Below, we provide a tighter bound on the complexity.

**Theorem 2** (Complexity). *The maximum number of recursions in Algorithm 1 is given by*

$$\sum_{v \in \overline{\mathcal{G}}_{\mathcal{D}}} \min \left( \left( |b_{\overline{\mathcal{G}}_{\mathcal{D}}}(v)| + \sum_{i=1}^{D-1} \sum_{v' \in \mathcal{N}_i(v)} (|b_{\overline{\mathcal{G}}_{\mathcal{D}}}(v')| - 1) \right), ND \right),$$

*where $\mathcal{N}_i(v)$ is the set of nodes reached within $i$ steps of recursion from $v$. The runtime complexity is thus, worst case, $\mathcal{O}(|V|ND)$, but can be significantly lower for graphs $\overline{\mathcal{G}}_{\mathcal{D}}$ with low binary degree.*

For any conjunction, Algorithm 1, in general, finds only a subset of its ground patterns in $\overline{\mathcal{G}}_{\mathcal{D}}$. Thus, the utility measures computed based on the mined patterns will be estimates of the actual values. We quantify these utility estimates by means of $\varepsilon$-*uncertainty*, in line with Feldstein et al. (2023b).

**Definition 9** ($\varepsilon$-uncertainty). *An estimate $\hat{s}$ of a scalar $s$ is $\varepsilon$-uncertain, $\varepsilon \in [0, 1)$, if $|\hat{s} - s|/s < \varepsilon$.*

**Definition 10** (Pattern occurrence distribution). *The pattern occurrence distribution $P_{\mathcal{D}}$, subject to $\mathcal{D}$, is the function that maps each connected pattern to the number of its groundings in $\mathcal{D}$.*

Theorem 3 provides a bound on the maximum number of paths $N$ needed by Algorithm 1 to guarantee $\varepsilon$-uncertainty utility estimates. This quantifies the trade-off between accuracy and runtime.

**Theorem 3** (Optimality). *Let $\rho$ be a set of rules whose patterns are of length $\leq 2D + 1$, where for each $\rho \in \rho$, patterns $\mathcal{G}_{\text{body}(\rho)}$, $\mathcal{G}_{\text{head}(\rho)}$, and $\mathcal{G}_{\text{body}(\rho) \wedge \text{head}(\rho)}$ are among the $M$ patterns with the highest number of groundings in $\overline{\mathcal{G}}_{\mathcal{D}}$. If $P_{\mathcal{D}}$ is Zipfian, then to ensure that $\mathsf{U}(\rho)$ is $\varepsilon$-uncertain, the upper bound on $N$ in Algorithm 1 scales as $N \propto \mathcal{O}\left(\frac{MD}{\varepsilon^2}\right)$.*

**Comparison to random walks.** Prior work mined *motifs*, objects similar to ground patterns, using random walks (Kok & Domingos, 2010; Feldstein et al., 2023b). Intuitively, Algorithm 1 can be seen as running random walks in parallel, while avoiding repeating the same walk twice and thereby wasting computational effort. Running $N$ random walks of length $D$ from $|V|$ nodes requires $|V|ND$ steps, which is the same as the *worst-case* computational cost of Algorithm 1 (Theorem 2). Another advantage of Algorithm 1 over random walks is that it results in more accurate utility estimates:

1. Random walks might backtrack, therefore, finding ground patterns of tautologies. Algorithm 1 avoids this issue by neglecting any previously encountered edge.

2. Random walks may revisit paths that have already been walked before. In contrast, in each call to NEXTSTEP, Algorithm 1 either visits a previously unvisited edge in the graph or it terminates. Therefore each new computation provides new information.

3. Random walks can miss ground patterns since they randomly sample a subset. In contrast, Algorithm 1 is guaranteed to mine all ground patterns involving nodes that are $N$-close to the source node $v_0$, while ground patterns that involve nodes that are not $N$-close are mined with a higher probability than with random walks (Theorem 1).

# 6 UTILITY-BASED STRUCTURE LEARNING

We now introduce our structure learning pipeline, which we call SPECTRUM (Structural Pattern Extraction and Cheap Tuning of Rules based on Utility Measure), presented in Algorithm 2. In summary, SPECTRUM begins by mining patterns, then checks each mined pattern whether it is a pattern of a "useful" rule, and finally sorts the useful rules in a greedy fashion.

## 6.1 RESTRICTIONS ON THE MINED RULES

As we stated in the introduction, SPECTRUM focuses on learning Datalog rules (as opposed to general first-order logic formulae). In addition, the algorithms restrict the shape of the mined rules:

(1) Algorithm 1 only mines patterns (and by extension rules) where each term occurs in at most two binary predicates and one unary predicate, except for the special case $\text{P}_1(X) \rightarrow \text{P}_2(X)$ (line 2 in Algorithm 1).

(2) Algorithm 2 restricts to rules that are body-connected and *term-constrained*. A rule is term-constrained if every term occurs in at least two atoms of the rule.

For example, the rule $\text{FRIENDS}(U_1, U_2) \rightarrow \text{LIKES}(U_2, I)$ is not term-constrained, since neither $U_1$ nor $I$ appear twice, but $\text{LIKES}(U_1, I) \wedge \text{FRIENDS}(U_1, U_2) \rightarrow \text{LIKES}(U_2, I)$ is term-constrained.

Restriction (1) helps to restrict the complexity of the framework. Additionally, we recommend setting the terminal depth $D$ in Algorithm 1 to a small number, as $N$ scales with a factor of $D$ (Theorem 3), and thus Algorithm 1 has complexity $\mathcal{O}(D^2)$. In our experiments (Section 7), we set $D = 3$ (i.e. a maximum of three binary predicates per rule). This is not a severe limitation, as we show empirically that many useful rules can be expressed within this restriction.

Restriction (2) ensures useful rules: term-constrainedness ensures each term is in at least one known atom, aiding link prediction, while body-connectedness is required for computing utility (Section 4).

## 6.2 ALGORITHM

SPECTRUM requires three parameters $M$, $\varepsilon$, and $D$: $M$ is the maximum number of rules of the final theory; $D$ sets a limit to the length of the mined rules as the pattern length is limited to $2D + 1$; $\varepsilon$ balances the trade-off between accuracy in the utility measures and computational effort. Given these parameters, SPECTRUM computes an optimal $N$ for pattern mining (Theorem 3), and, using Algorithm 1, mines patterns which are stored in a map $\overline{\mathcal{G}}_{\text{global}}$ of patterns to their groundings in $\overline{\mathcal{G}}_{\mathcal{D}}$.

Each pattern $\mathcal{G}_\varphi$ in the keys of the map $\overline{\mathcal{G}}_{\text{global}}$ is considered in turn. Each rule that could have resulted in this pattern, i.e. a rule from the set $\mathcal{R} := \{\rho \mid \mathcal{G}_{\text{body}(\rho) \wedge \text{head}(\rho)} = \mathcal{G}_\varphi\}$, is considered. If a rule $\rho \in \mathcal{R}$ is term constrained and satisfies $\frac{\text{P}(\rho)\text{S}(\rho)}{\text{B}(\rho)} > 1$, i.e. the rule is a better predictor than a random guess (Section 4), then $\rho$ is added to the set of candidate rules $\rho_{\text{candidates}}$.

From the set of candidate rules, a subset of $M$ rules with the highest individual utility is chosen (Definition 6). The utility of each rule $\rho$ is directly calculated from $\overline{\mathcal{G}}_{\text{global}}$; since this is a map from patterns $\mathcal{G}_\varphi$ of a conjunction $\varphi$ to its groundings in $\overline{\mathcal{G}}_{\mathcal{D}}$, i.e. $\overline{\mathcal{G}}_\varphi$. The quantity $|\overline{\mathcal{G}}_\varphi|$ can be looked up in the map. The conjunction $\varphi$ can be $\text{body}(\rho)$, $\text{head}(\rho)$, or $\text{body}(\rho) \wedge \text{head}(\rho)$. The complexity and symmetry factor can be computed for each $\rho$ directly from its length, rule pattern and body pattern.

---

**Algorithm 2:** SPECTRUM

---

**Input:** $\mathcal{D}$ – relational database
**Output:** $\boldsymbol{\rho}$ – set of rules ordered by utility
**Parameters:** $M$ – the number of top patterns to consider as rules
            $\varepsilon$ – target uncertainty of the utility estimates
            $D$ – maximum depth of pattern mining

1   $N \leftarrow$ COMPUTE_OPTIMAL_N$(M, \varepsilon, D)$                       `// Thm. 3`
2   $\overline{\mathcal{G}}_{\text{global}} \leftarrow$ PATTERNMINING$(\overline{\mathcal{G}}_{\mathcal{D}}, N, D)$                    `// Alg. 1`
3   $\boldsymbol{\rho}_{\text{candidates}} \leftarrow \emptyset$
4   **for each** $\mathcal{G}_{\varphi}$ **in** $\overline{\mathcal{G}}_{\text{global}}$ **do**
5      **for each** $\rho$ **in** $\mathcal{R} := \{\rho \mid \mathcal{G}_{\text{body}(\rho) \wedge \text{head}(\rho)} = \mathcal{G}_{\varphi}\}$ **do**
6          **if** $\rho$ is body-connected and term-constrained and $\frac{\mathsf{P}(\rho) \cdot \mathsf{S}(\rho)}{\mathsf{B}(\rho)} > 1$ **then**
7             $\boldsymbol{\rho}_{\text{candidates}} \leftarrow \boldsymbol{\rho}_{\text{candidates}} \cup \{\rho\}$

8   $\boldsymbol{\rho}_{\text{candidates}} \leftarrow$ CHOOSETOP_$M(\boldsymbol{\rho}_{\text{candidates}}, M)$     `// Ranked by individual utility`
9   $\boldsymbol{\rho}_{\text{final}} \leftarrow [\,]$                                    `// Initialise an empty vector`
10   **while** $\boldsymbol{\rho}_{\text{candidates}}$ **is not** empty **do**        `// Order rules by contributed utility`
11      $\rho_{\text{best}} \leftarrow \emptyset$
12      **for each** $\rho$ **in** $\boldsymbol{\rho}_{\text{candidates}}$ **do**
13          **if** $\mathsf{U}(\{\rho\} \cup \boldsymbol{\rho}_{\text{final}}) > \mathsf{U}(\{\rho_{\text{best}}\} \cup \boldsymbol{\rho}_{\text{final}})$ **then**
14             $\rho_{\text{best}} \leftarrow \rho$

15      $\boldsymbol{\rho}_{\text{candidates}} \leftarrow \boldsymbol{\rho}_{\text{candidates}} \setminus \{\rho_{\text{best}}\}$
16      **append** $\rho_{\text{best}}$ to $\boldsymbol{\rho}_{\text{final}}$

17   **return** $\boldsymbol{\rho}_{\text{final}}$

---

SPECTRUM then orders the remaining $M$ rules in order of their contribution to the theory utility (Definition 8). The algorithm starts by finding the rule with the highest utility and stores it in a vector $\boldsymbol{\rho}_{\text{final}}$. Then, in each iteration of the while-loop, SPECTRUM finds the rule out of the remaining ones that provides the highest increase in theory utility when added to the current rules in $\boldsymbol{\rho}_{\text{final}}$.

After SPECTRUM, the rules $\boldsymbol{\rho}_{\text{final}}$ can be passed to any probabilistic logical framework (e.g. PSL or MLN) to learn the weights of the rules (i.e. the likelihood of the rule being satisfied) for a given dataset $\mathcal{D}$. We recommend adding one rule at a time to the logical model (in the order they were added to the vector $\boldsymbol{\rho}_{\text{final}}$) when learning the weights. One can then validate the different theories by checking when the accuracy drops, as more rules may not imply a better theory.

# 7   EXPERIMENTS

We conduct three experiments. First, we compare SPECTRUM to state-of-the-art MLN structure learners, achieving a 16% accuracy improvement and reducing runtime to under 1%. However, as current MLN implementations struggle with large datasets—and our primary objective is to showcase SPECTRUM's scalability—we defer the discussion of these experiments and additional details to Appendix D. Second, we demonstrate the scalability of SPECTRUM for learning PSL models on datasets used by neuro-symbolic frameworks (Section 7.1). Third, we benchmark SPECTRUM on knowledge graph completion against leading neural network approaches (Section 7.2).

## 7.1   SCALABLE LEARNING OF PROBABILISTIC LOGICAL MODELS

**Task.** For each dataset, our goal is to learn PSL rules that are the same (or better) than the hand-engineered ones. For Citeseer, Cora and Yelp, hand-engineered rules are provided by Bach et al. (2017). For CAD, they are provided by London et al. (2013). Note that the baselines used to evaluate SPECTRUM on MLNs do not scale to datasets of the size considered here, which is why we compare against the hand-engineered logical theories.

**Results.** For Citeseer, Cora and CAD, SPECTRUM recovers all hand-crafted rules. For Yelp, SPECTRUM recovers all hand-crafted rules that are of a form learnable by SPECTRUM (2 rules are not, because they are not term-constrained). See Appendix D for full details of the learnt rules for each dataset. We report the structure learning times for each dataset in Table 1. As expected, the runtime increases roughly linearly with the dataset size. Note that for the CAD experiments, we used $M = 60$ instead of $M = 30$ because of the proportionally larger number of different predicates in that dataset. Importantly, SPECTRUM can process $\sim 10^6$ facts in the same time that PRISM and LSM can process only $\sim 10^3$ facts (Table 4), demonstrating how SPECTRUM successfully overcomes the scalability issues of prior art.

Table 1: Comparison of runtimes and fraction of rules recovered across datasets of varying size.

|  | Citeseer | Cora | CAD | Yelp |
|---|---|---|---|---|
| Dataset Size | $6.8 \times 10^3$ | $6.9 \times 10^3$ | $2.5 \times 10^5$ | $2.2 \times 10^6$ |
| Training Time / s | $2.08 \pm 0.02$ | $2.44 \pm 0.02$ | $84 \pm 0.5$ | $348 \pm 2$ |
| Rules recovered | 7/7 | 6/6 | 21/21 | 24/26 |

## 7.2 KNOWLEDGE GRAPH COMPLETION

**Task.** Knowledge completion is a task commonly used by neural network approaches to structure learning to assess the quality of the learnt rules e.g. as in NeuralLP (Yang et al., 2017) and DRUM (Sadeghian et al., 2019). In contrast to the previous experiments, where the goal is to predict entire facts (i.e. P$(X, Y)$), here, the goal is only to infer missing entities (i.e. given P(alice, $X$) predict $X$).

**Results.** For evaluation, we used the NCRL script (Cheng et al., 2023) and report three evaluation metrics, namely Mean Reciprocal Rank (MRR), Hit at 1 and Hit at 10. Since NCRL does not provide a method for learning rule weights, we used our precision metric instead when evaluating SPECTRUM. Predicted entities are ranked by summing up the confidence values of every rule that is satisfied with that entity in its grounding. We compare SPECTRUM against three SOTA methods – AMIE3 (Lajus et al., 2020), RNNlogic (Qu et al., 2021a), and NCRL (Cheng et al., 2023) – on five widely used benchmark datasets: Family (Hinton, 1986), UMLS (Kok & Domingos, 2007), Kinship (Kok & Domingos, 2007), WN18RR (Dettmers et al., 2018) and FB15K-237 (Toutanova & Chen, 2015). Evaluation results are shown in Table 2 and dataset statistics are shown in Appendix D. RNNLogic and NCRL experiments ran on V100 GPUs. SPECTRUM ran on a 12-core 2.60GHz i7-10750H CPU.

Table 2: Comparison of runtime (s) and evaluation metrics for RNNLogic, NCRL, and SPECTRUM. Dashed lines indicate a timeout ($> 10$h), and slashes denote failure due to insufficient memory.

|  | AMIE3 | | | RNNLogic | | | NCRL | | | SPECTRUM | | |
|---|---|---|---|---|---|---|---|---|---|---|---|---|
|  | Time | MRR | Hit10 | Time | MRR | Hit10 | Time | MRR | Hit10 | Time | MRR | Hit10 |
| Family | 4.8 | 0.430 | 0.766 | 1200 | 0.278 | 0.494 | 88 | 0.873 | 0.993 | **1.5** | **0.920** | **1.00** |
| UMLS | 204 | 0.064 | 0.161 | 1200 | 0.689 | 0.824 | 420 | 0.659 | 0.853 | **2.9** | **0.759** | **0.935** |
| Kinship | 884 | 0.168 | 0.454 | 1300 | 0.535 | 0.919 | 480 | **0.592** | **0.897** | **3.9** | 0.500 | 0.892 |
| WN18RR | **3.9** | 0.079 | 0.087 | – | – | – | 2700 | 0.506 | 0.687 | 36 | **0.530** | **0.900** |
| FB15K-237 | 171 | 0.136 | 0.239 | – | – | – | / | / | / | 260 | **0.304** | **0.462** |

**Discussion.** With a few exceptions (notably the Kinship dataset), SPECTRUM outperforms neural network methods in MRR, Hit1, and Hit10, while consistently offering a significantly faster runtime ($\sim 100$x improvement) and more efficient memory usage. Note that the results in Table 2 were obtained using neural network experiments on a server with a V100 GPU (30Gb memory, 40 CPUs), while initial attempts on a 6Gb GPU failed. In contrast, the results reported for SPECTRUM were obtained on a laptop with 12 CPUs. The runtime for NCRL excludes the additional cost of hyperparameter tuning (6 hyperparameters need tuning), while SPECTRUM was run with fixed hyperparameters (fixed $M = 20 \times$ [number of relations] and fixed $\varepsilon = 0.01$). The poor memory scaling of NCRL meant that it ran out of memory even on a relatively small dataset (the

FB15K-237 dataset with $\sim 10^4$ facts). This implies that NCRL would not scale to the CAD ($\sim 10^5$) or Yelp ($\sim 10^6$) dataset in Section 7.1. Finally, SPECTRUM offers a key advantage when it comes to explainability – it orders these rules by their contribution to the theory utility.

## 8 RELATED WORK

**Markov logic networks.** State-of-the-art structure learning approaches proceed in two steps - identification of patterns and optimisation. The different structure learning techniques for MLNs can be split into two groups: (1) methods for which patterns are user-defined and (2) methods for which patterns are identified automatically in the data. The current state-of-the-art that does not require user-defined patterns is LSM (Kok & Domingos, 2010). LSM identifies patterns by running random walks over a hypergraph representation of the data. A major limitation of LSM is that it lacks guarantees on the quality of the mined patterns. PRISM is an efficient pattern-mining technique with theoretical guarantees on the quality of mined patterns, solving the above limitation (Feldstein et al., 2023b). Our empirical results show that SPECTRUM scales significantly better than any of the techniques mentioned above, without requiring any domain expertise as patterns are mined automatically. In addition, inspired by Feldstein et al. (2023b), we provide $\varepsilon$-uncertainty guarantees, which, in contrast to PRISM, are guarantees on the utility of the output theory rather than just the patterns.

**Inductive logic programming.** A popular family of techniques for learning Datalog theories is *inductive logic programming* (ILP), e.g. FOIL (Quinlan, 1990), *MDIE* (Muggleton, 1995) and *Inspire* (Schüller & Benz, 2018). Given a database $\mathcal{D}$, a set of positive facts $E^+$ and a set of negative facts $E^-$, ILP-based techniques work by computing a theory that along with $\mathcal{D}$ entails all facts in $E^+$ and no fact in $E^-$. MetaAbd (Dai & Muggleton, 2021) mixes logical abduction (i.e., backward reasoning) (Kakas, 2017) with ILP to simultaneously learn a logical theory and train a neural classifier. A recent line of research proposed formulations of ILP in which part of the computation can be differentiated and, hence, (part of) the learning is done through backpropagation. For example, Evans & Grefenstette (2018) introduced $\delta$ILP which employs the semantics of fuzzy logic for interpreting rules. Sen et al. (2022) proposed a similar ILP technique based on logical neural networks (Riegel et al., 2020). All ILP-based techniques require users to provide the patterns of the formulae to be mined. The above requirement, along with issues regarding scalability (Evans & Grefenstette, 2018), limits the applicability of ILP techniques in large and complex datasets.

**Differentiable rule learning.** Other techniques for learning logical rules in a differentiable fashion are *NeuralLP* (Yang et al., 2017), *DRUM* (Sadeghian et al., 2019), *neural logic machines* (NLMs) (Dong et al., 2019), *RNNLogic* (Qu et al., 2021b), and *NCRL* (Cheng et al., 2023). All these techniques are limited by the shape of the rules that they learn. NeuralLP, DRUM, RNNLogic and NCRL can only learn rules of the form $P_1(X, Z_1) \wedge \cdots \wedge P_{n-1}(Z_{n-1}, Y) \rightarrow P_n(X, Y)$, while NLMs are restricted to learning rules where all head and body atoms contain the same variables. While SPECTRUM has some limitation on the shapes of the rules it learns, they are less restrictive.

## 9 CONCLUSIONS

A major point of criticism against neurosymbolic techniques and logical models is the need for domain expertise (Feldstein et al., 2023a; Huang et al., 2021; Li et al., 2023). This work tackles the scalability issue of learning logical models from data, mining accurate logical theories in minutes for datasets with millions of instances, thus making the development of a logical model a simple and fast process. Therefore, we see our work as having the potential to increase the adoption of neurosymbolic frameworks. In addition, learning logical models improves explainability by extracting knowledge from data that is interpretable by a domain expert.

There are several directions for future research. First, the pattern mining algorithm could be generalised to relations with higher arity. Second, the pattern mining algorithm could be extended to require fewer restrictions on the shape of the rules. Finally, since the rules we learn are model agnostic, we plan to apply our technique to other logical frameworks, in addition to MLNs and PSL, such as Problog (De Raedt et al., 2007).

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

## A  PROOF OF THEORETICAL PROPERTIES OF PATTERN MINING

In this section, we prove Theorems 1, 2, and 3 and justify Remark 2. We start with some preliminary definitions before proceeding with the proof.

**Definitions and notation.**  The *D-neighbourhood of a node* $v_i$ is the set of all nodes that are a distance less than or equal to $D$ from $v_i$. The *D-neighbourhood length $l$ patterns of a node* $v_i$, denoted $\mathcal{P}_{D,l}(v_i)$, is the set of all connected patterns of length $l$ that have a grounding that includes $v_i$, and whose remaining nodes in the grounding also occur within the $D$-neighbourhood of $v_i$. The *D-neighbourhood length $l$ pattern distribution of* $v_i$ is the function that maps from $\mathcal{P}_{D,l}(v_i)$ to the number of groundings of that pattern within the $D$-neighbourhood of $v_i$ that include $v_i$. The *D-neighbourhood length $l$ pattern probability distribution of* $v_i$, denoted $P_i^{(l)}$, is the probability distribution obtained by normalising this distribution.

## A.1 Proof of Theorem 1

*Proof.* We prove this theorem by proving the two statements individually:

**S1:** Algorithm 1 finds all patterns of length $l \leq D$ that only involve the source node and other $N$-close nodes.

**S2:** For patterns involving nodes that are not $N$-close, Algorithm 1 finds them with a probability larger than when running random walks.

**Statement 1:** Consider a node $v'$ that is a distance $l \leq D$ away from a source node $v_0 = v_{i_0}$ and consider a generic path of length $l$ from $v_{i_0}$ to $v' = v_{i_l}$, $(v_{i_0}, e_{i_0}, \ldots, e_{i_{l-1}}, v_{i_l})$. First, notice that if $N \geq |\mathcal{E}'| = |\mathsf{b}_{\overline{\mathcal{G}}_\mathcal{D}}(v_{i_0})|$, then the edge $e_{i_0}$ will certainly be discovered by Algorithm 1 (selection step, lines 11-16). Similarly, for the second edge $e_{i_1}$ to be found, we need the value of $n$ upon reaching node $v_{i_1}$ to satisfy $n \geq |\mathcal{E}'|$ (selection step, lines 11-16). Note also that $|\mathcal{E}'| \leq |\mathsf{b}_{\overline{\mathcal{G}}_\mathcal{D}}(v_{i_1})| - 1$, since the previous incident edge to $v_{i_1}$ is excluded from the set $\mathcal{E}'$ (line 11). Therefore, a sufficient condition for the edge $e_{i_1}$ to be included is $N \geq |\mathsf{b}_{\overline{\mathcal{G}}_\mathcal{D}}(v_{i_0})|(|\mathsf{b}_{\overline{\mathcal{G}}_\mathcal{D}}(v_{i_1})| - 1)$. Reasoning inductively, a sufficient condition for every edge in the path to be found by Algorithm 1 is

$$N \geq |\mathsf{b}_{\overline{\mathcal{G}}_\mathcal{D}}(v_{i_0})| \prod_{j=1}^{l-1}(|\mathsf{b}_{\overline{\mathcal{G}}_\mathcal{D}}(v_{i_j})| - 1). \tag{1}$$

If this holds for all possible length $l$ paths between $v_{i_0}$ and $v_{i_l}$, then all of those paths will be discovered. This is equivalent to the statement that $v_{i_l}$ is $N$-close to $v_{i_0}$. Therefore, for any node that is $N$-close to $v_{i_0}$ (and is a distance $l$ from $v_{i_0}$), all possible length $l$ paths leading to that node will be found. Since any connected patterns is a subsets of a path, all possible patterns of length $l \in \{1, 2, \ldots, D\}$ have been found by Algorithm 1, thus completing the proof.

**Statement 2:** First, notice that if a node is not $N$-close, then there is still a chance that it could be found due to random selection. This is because the smallest value of $n$ is 1 and if $n < |\mathcal{E}'|$ then we proceed by choosing the next edge in the path uniform randomly (line 13). Worst-case, $\mathsf{b}_{\overline{\mathcal{G}}_\mathcal{D}}(v_{i_0})| \geq N$, in which case the algorithm runs $N$ different paths where each edge is chosen at random. This is almost equivalent to running $N$ random walks, with the difference that Algorithm 1 does not allow backtracking or visiting previously encountered nodes, which increases the probability of finding novel nodes compared to independent random walks. Thus, the probability that a node is found using Algorithm 1 is strictly larger than when running $N$ independent random walks from $v_0$. $\square$

## A.2 Proof of Theorem 2

*Proof.* We prove this by partitioning the possibilities into three cases and proving that the upper bound formula is true in all cases.

**Case 1 – every node in the $D$-neighbourhood of $v_0$ is $N$-close to $v_0$:** In this case, the number of calls to NextStep is given by the total number of selections of binary edges (i.e. the cumulative sum of $|\mathcal{E}'|$ every time it is computed). Recall, from the discussion in the proof of Theorem 1, that for node $v_0$, $|\mathcal{E}'| \leq |\mathsf{b}_{\overline{\mathcal{G}}_\mathcal{D}}(v_0)|$, and for all other nodes $v_i$, $|\mathcal{E}'| \leq |\mathsf{b}_{\overline{\mathcal{G}}_\mathcal{D}}(v_i)| - 1$. Therefore, the sum of $|\mathcal{E}'|$ is upper bounded by the sum of the RHSs of these inequalities. Summing over all nodes gives the quantity in the left-hand argument of the minimum function in Theorem 2[2]. Note that, in this case, the quantity in the right-hand argument, $ND$, is strictly larger since this is the maximum computation when running $N$ paths of length $D$ without avoiding previously encountered edges. The minimum therefore gives a valid upper bound.

**Case 2 – no node (other than the source node) exists in the $D$-neighbourhood of $v_0$ that is $N$-close to $v_0$:** In this case, $N$ recursions are called in the first step, and each following recursion will call one recursion until the final depth $D$ is reached, totalling $ND$ recursions, which is the right-hand argument of the minimum function in Theorem 2. In this case the LHS is actually larger

---

[2]In practice, the true number of recursions is likely to be considerably less than this, due to the avoidance of previously explored edges when passing binaries onto the next step (line 11).

than the RHS, since the branching factor of paths is strictly larger at the first step. The minimum therefore gives a valid upper bound.

**Case 3 – some nodes are $N$-close and other nodes are not:** In this case, the number of recursions is strictly less than the left-hand argument of the minimum function in Theorem 2, for if it wasn't, then by definition every node would be $N$-close (contradiction). Likewise, it is also strictly less than $ND$, since this is the maximum computation when running $N$ paths of length $D$ without avoiding previously encountered edges. The minimum of the two is therefore also a valid upper bound.

Therefore, in all possible cases, the minimum of these two quantities gives a valid upper bound for the number of recursions, and thus the computational complexity, of Algorithm 1.  $\square$

### A.3  PROOF OF THEOREM 3

*Proof.* We will prove the theorem in three stages:

**S1:** We derive an upper bound, $N(\varepsilon')$, on the number of purely random walks required to achieve $\varepsilon'$-uncertainty of the top $M$ pattern probabilities;

**S2:** We derive the corresponding $N(\varepsilon)$ required to achieve $\varepsilon$-uncertainty in the utility of an arbitrary set of rules whose head pattern, body pattern and rule patterns belong to these top $M$ patterns, under purely random walks;

**S3:** We prove that running Algorithm 1 with $N = N(\varepsilon)$ leads to a strictly lower $\varepsilon$-uncertainty for this rule utility than when using purely random walks, thus $N(\varepsilon)$ satisfies the theorem claim.

*Stage 1 of the proof is an adaptation of a similar proof for $\varepsilon$-uncertainty of path probabilities of random walks on hypergraphs by Feldstein et al. Feldstein et al. (2023b).*

**Stage 1** *Throughout this proof, we will consider pattern probabilities within the $D$-neighbourhood of nodes, where $D$ is fixed by Algorithm 1.*

Given a node $v_i \in \mathcal{D}_G$, let $P_i^{(l)}(\mathcal{G}_k)$ denote the pattern probability of the $k^{th}$ most common pattern in the $D$-neighbourhood length $l$ patterns of $v_i$ (note that the constraints we make on rule patterns in Section 6.1 means that we can bound $l \le 2D + 1$, where $l$ can exceed $D$ due to the presence of unary predicates in the rule pattern). The Ziphian assumption implies that

$$P_i^{(l)}(\mathcal{G}_k) = \frac{1}{kZ}, \tag{2}$$

where $Z = \sum_{k=1}^{|\mathcal{P}_{D,l}(v_i)|} \frac{1}{k}$ is the normalisation constant.

Consider running $N$ random walks from $v_i$ without backtracking, and up to a maximum depth of $D$ (c.f. Algorithm 1). Since the walks are uniform random, a partial walk up to step $l \le 2D + 1$ yields a random sample from the $D$-neighbourhood length $l$ pattern probability distribution of $v_i$. Denote by $\hat{C}_{i,N}^{(l)}(\mathcal{G}_k)$ the number of times that the $k^{th}$ most probable pattern, $\mathcal{G}_k$, was sampled after running all $N$ random walks. By independence of the random walks, the quantity $\hat{C}_{i,N}^{(l)}(\mathcal{G}_k)$ is a binomially distributed random variable with

$$\mathbb{E}\left[\hat{C}_{i,N}^{(l)}(\mathcal{G}_k)\right] = N P_i^{(l)}(\mathcal{G}_k); \quad \mathrm{Var}\left[\hat{C}_{i,N}^{(l)}(\mathcal{G}_k)\right] = N P_i^{(l)}(\mathcal{G}_k)(1 - P_i^{(l)}(\mathcal{G}_k)).$$

It follows that the pattern probability estimate $\hat{P}_i^N(\mathcal{G}_k^{(l)}) := \hat{C}_{i,N}^{(l)}(\mathcal{G}_k)/N$ has fractional uncertainty $\epsilon(\mathcal{G}_k)$ given by

$$\epsilon\left(\mathcal{G}_k\right) := \frac{\sqrt{\mathrm{Var}\left[\hat{P}_i^N(\mathcal{G}_k^{(l)})\right]}}{\mathbb{E}\left[\hat{P}_i^N(\mathcal{G}_k^{(l)})\right]} = \sqrt{\frac{1 - P_i^{(l)}(\mathcal{G}_k)}{N P_i^{(l)}(\mathcal{G}_k)}}$$

$$= \sqrt{\frac{k\left(\sum_{m=1}^{|\mathcal{P}_{D,l}(v_i)|} \frac{1}{m}\right) - 1}{N}}, \tag{3}$$

where in the second line we used the Ziphian assumption equation 2. Suppose further that we require that all pattern probabilities $P_i^{(l)}(\mathcal{G}_k)$ up to the $M^{\text{th}}$ highest probability for that length have $\varepsilon'$-uncertainty, i.e.

$$\varepsilon' = \max_{k \in \{1,2,\ldots,M\}} \epsilon(\mathcal{G}_k) = \epsilon(\mathcal{G}_M),$$

where $\mathcal{G}_M$ is the $M^{\text{th}}$ most probable pattern, and so, upon rearranging,

$$N(\varepsilon') = \frac{M\left(\sum_{m=1}^{|\mathcal{P}_{D,l}(v_i)|} \frac{1}{m}\right) - 1}{\varepsilon'^2}. \tag{4}$$

We have

$$N(\varepsilon') \approx \frac{M\left(\gamma + \ln(|\mathcal{P}_{D,l}(v_i)|)\right)}{\varepsilon'^2}, \tag{5}$$

where we used the log-integral approximation for the sum of harmonic numbers $\sum_{m=1}^{|\mathcal{P}_{D,l}(v_i)|} \frac{1}{m} = \gamma + \ln(|\mathcal{P}_{D,l}(v_i)|) + \mathcal{O}\left(\frac{1}{|\mathcal{P}_{D,l}(v_i)|}\right)$, where $\gamma \approx 0.577$ is the Euler-Mascheroni constant. Equation equation 5 gives an upper bound on the number of random walks required to achieve $\varepsilon'$-uncertainty of the top $M$ most common pattern probabilities of length $l$ that occur in the 3-neighbourhood of node $v_i$. Note that the exact value of $|\mathcal{P}_{D,l}(v_i)|$ depends on the specifics of the dataset, however, in general, it would grow exponentially with the length $l$ due to a combinatorial explosion in the number of patterns Feldstein et al. (2023b). This means that $N(\varepsilon')$ scales as

$$N(\varepsilon') \sim \mathcal{O}\left(\frac{Ml}{\varepsilon'^2}\right).$$

If we want to ensure $\varepsilon'$ uncertainty for patterns of all lengths $l \in \{1, 2, \ldots, 2D+1\}$ then we conclude that $N(\varepsilon')$ should scale as

$$N(\varepsilon') \sim \mathcal{O}\left(\frac{MD}{\varepsilon'^2}\right).$$

This concludes stage 1.

**Stage 2** Assuming that the top $M$ most common pattern probabilities of length $l$ are $\varepsilon'$-uncertain, for all $l \in \{1, 2, \ldots, 2D+1\}$, we now derive an upper bound for the level of uncertainty of the utility of an arbitrary set of rules whose head patterns, body patterns and rule patterns belong to these top $M$ patterns.

Recall that the precision of a rule $\rho$ can be expressed as the ratio of the number of groundings of $\text{head}(\rho) \wedge \text{body}(\rho)$, to the number of groundings of $\text{body}(\rho)$ in the data i.e.

$$\mathsf{P}(\rho) = \frac{|\overline{\mathcal{G}}_{\text{body}(\rho) \wedge \text{head}(\rho)}|}{|\overline{\mathcal{G}}_{\text{body}(\rho)}|}.$$

Computing precision exactly would require exhaustively sampling the entire dataset. However, we can still obtain an unbiased estimate of precision, $\widehat{\mathsf{P}}(\rho)$, using the ratio of counts of these ground patterns from running random walks. Assuming that $\mathcal{G}_{\text{body}(\rho) \wedge \text{head}(\rho)}$ is a length $l$ pattern:

$$\widehat{\mathsf{P}}(\rho) = \frac{\sum_{v_i \in \overline{\mathcal{G}}_D} \hat{C}_{i,N}^{(l)}\left(\mathcal{G}_{\text{body}(\rho) \wedge \text{head}(\rho)}\right)}{\sum_{v_i \in \overline{\mathcal{G}}_D} \hat{C}_{i,N}^{(l)}\left(\mathcal{G}_{\text{body}(\rho)}\right)},$$

is an unbiased estimator for $\mathsf{P}(\rho)$. Assuming the rule's head, body and rule patterns belong to the top $M$ patterns, then we know that the numerator and denominator both individually have $\varepsilon'$-uncertainty so we have, in the worst case, that $\widehat{\mathsf{P}}(\rho)$ has $\varepsilon$-uncertainty where $\varepsilon = 2\varepsilon'$.

Next, we consider the estimate of the quantity $|\overline{\mathcal{G}}_{\mathsf{body}(\rho)\wedge\mathsf{head}(\rho)}^{\mathsf{head}(\rho)=\overline{\alpha}}|$, where $\overline{\alpha}$ is a grounding of the head of the rule in the data. For brevity, we call the size of this set the *recall degree* of $\overline{\alpha}$ given a rule, and denote it as $\mathsf{D}_\rho(\overline{\alpha}) := |\overline{\mathcal{G}}_{\mathsf{body}(\rho)\wedge\mathsf{head}(\rho)}^{\mathsf{head}(\rho)=\overline{\alpha}}|$. Consider an arbitrary fact $\overline{\alpha}$ in the data that is in the $D$-neighbourhood of node $v_i$, and is the head predicate of a rule whose pattern $\mathcal{G}_{\mathsf{body}(\rho)\wedge\mathsf{head}(\rho)}$ can be traversed, without backtracking, starting from $v_i$. The probability, $q$, that $\overline{\alpha}$ is *not* discovered as part of that rule after $N(\varepsilon')$ random walks is given by

$$q = (1 - p')^{N(\varepsilon')},$$

where in the above, $p'$ is shorthand for $P_i^{(l)}(\mathcal{G}_{\mathsf{body}(\rho)\wedge\mathsf{head}(\rho)})$. We have

$$\ln(q) = N(\varepsilon')\ln(1 - p') < -N(\varepsilon')p'$$

and hence

$$q < e^{-N(\varepsilon')p'}.$$

But since, by the Ziphian assumption, $p' > \frac{1}{Z \cdot M} \approx \frac{1}{M(\gamma + \ln P)}$, we have that $p'N(\varepsilon') > \frac{1}{\varepsilon'^2}$ and hence

$$q < e^{-\frac{1}{\varepsilon'^2}}.$$

The above inequality holds for arbitrary $v_i$, hence the expectation of the estimated recall degree satisfies

$$\mathsf{D}_\rho(\overline{\alpha}) > \mathbb{E}[\widehat{\mathcal{D}}_\rho(\overline{\alpha})] > \mathsf{D}_\rho(\overline{\alpha})(1 - e^{-\frac{1}{\varepsilon'^2}}),$$

where $\mathsf{D}_\rho(\overline{\alpha})$ is the true recall degree of $\overline{\alpha}$. Since $\varepsilon' > e^{-\frac{1}{\varepsilon'^2}}$ for all $0 < \varepsilon' < 1$, we conclude that $\widehat{\mathcal{D}}_\rho(\overline{\alpha})$ has $\varepsilon'$-uncertainty. Therefore, by the Taylor expansion, the estimated log-recall, $\ln(1 + \widehat{\mathcal{D}}_\rho(\overline{\alpha}))$ also has $\varepsilon'$ uncertainty, as does the estimated rule-set log-recall $\mathsf{R}(\boldsymbol{\rho}_\alpha) = \ln\left(1 + \sum_{\rho\in\boldsymbol{\rho}_\alpha}\widehat{\mathcal{D}}_\rho(\overline{\alpha})\right)$.

Note that the symmetry factor $\mathsf{S}(\rho)$ is known exactly for every rule, as it is a topological property of the rule rather than a property of the data. For the same reason, the complexity factor $\mathsf{C}(\rho)$ is also known exactly. Finally, the Bayesian prior $\mathsf{B}(\rho)$ is also known exactly, since computing it requires summing over the data once, which we do once at the beginning of SPECTRUM, and this only takes linear time.

Using the above results, we conclude that the rule-set utility

$$\mathsf{U}(\boldsymbol{\rho}) = \sum_{\alpha\in\boldsymbol{\alpha}}\left(\sum_{\rho\in\boldsymbol{\rho}_\alpha}\frac{\mathsf{P}(\rho)\mathsf{S}(\rho)}{\mathsf{B}(\rho)}\right) \cdot \mathsf{R}(\boldsymbol{\rho}_\alpha)\mathsf{C}(\boldsymbol{\rho}_\alpha),$$

has, by error propagation, worst case $\varepsilon$-uncertainty with $\varepsilon = 3\varepsilon'$.

Substituting $\varepsilon' = \varepsilon/3$ into equation 5, we conclude that an upper bound on the number of random walks required to guarantee $\varepsilon$-uncertainty of the rule-set utility, $\mathsf{U}(\boldsymbol{\rho})$ (where all rules' head patterns, body patterns and rule patterns belong to the $M$ most common patterns of their respective length) under random walks is given by

$$N(\varepsilon) = \frac{9M\left(\gamma + \ln(|\mathcal{P}_{D,l}(v_i)|)\right)}{\varepsilon^2}. \tag{6}$$

Considering all patterns of length $l \in \{1, 2, \ldots, 2D+1\}$ we see that this scales as

$$N(\varepsilon) \sim \mathcal{O}\left(\frac{MD}{\varepsilon^2}\right).$$

This concludes stage 2.

**Stage 3** We consider now the Algorithm 1. Let $v_i$ denote the source node of a fragment mining run. Set $N = N(\varepsilon)$. Partition the $D$-neighbourhood of $v_i$ into two sets, $\mathcal{N}_i^{\mathsf{close}}$ and $\mathcal{N}_i^{\mathsf{far}}$ - nodes that are

$N(\varepsilon)$-close and not $N(\varepsilon)$-close to $v_i$ respectively (Theorem 1). By the definition of $N$-close, setting $N = N(\varepsilon')$ in Algorithm 1 guarantees that all patterns containing nodes exclusively with $\mathcal{N}_i^{\text{close}}$ are counted exactly, whereas patterns that contain nodes within $\mathcal{N}_i^{\text{far}}$ are, in the worst case *not* counted with a probability given by

$$q = (1 - p')^{N(\varepsilon')},$$

with $p' = P_i^{(l)}(\mathcal{G}_{\text{body}(\rho)\wedge\text{head}(\rho)})$, assuming that $\rho$ is a length $l$ rule.

Partitioning $\hat{C}_{i,N}^{(l)}\left(\mathcal{G}_{\text{body}(\rho)\wedge\text{head}(\rho)}\right)$ into *close* and *far* contributions, we can write

$$\hat{C}_{i,N}^{(l)}\left(\mathcal{G}_{\text{body}(\rho)\wedge\text{head}(\rho)}\right) = \hat{C}_{i,\text{close},N}^{(l)}\left(\mathcal{G}_{\text{body}(\rho)\wedge\text{head}(\rho)}\right) + \hat{C}_{i,\text{far},N}^{(l)}\left(\mathcal{G}_{\text{body}(\rho)\wedge\text{head}(\rho)}\right).$$

In the above, by $\hat{C}_{i,\text{close},N}^{(l)}\left(\mathcal{G}_{\text{body}(\rho)\wedge\text{head}(\rho)}\right)$ we mean the number of times the pattern $\mathcal{G}_{\text{body}(\rho)\wedge\text{head}(\rho)}$ was counted with nodes that are exclusively in the set $\mathcal{N}_i^{\text{close}}$. Similarly, $\hat{C}_{i,\text{far},N}^{(l)}\left(\mathcal{G}_{\text{body}(\rho)\wedge\text{head}(\rho)}\right)$ is the number of times that $\mathcal{G}_{\text{body}(\rho)\wedge\text{head}(\rho)}$ was counted with nodes that are in a mixture of $\mathcal{N}_i^{\text{close}}$ and $\mathcal{N}_i^{\text{far}}$. Note that $\hat{C}_{i,\text{close},N}^{(l)}\left(\mathcal{G}_{\text{body}(\rho)\wedge\text{head}(\rho)}\right)$ is an exact count and has no uncertainty due to the exhaustive property of Algorithm 1 for $N$-close nodes.

Using the result from stage 2 of the proof, we know that $q < \varepsilon'$ and hence $\hat{C}_{i,\text{far},N}^{(l)}\left(\mathcal{G}_{\text{body}(\rho)\wedge\text{head}(\rho)}\right)$ has, worst case, $\varepsilon'$-uncertainty and so $\hat{C}_{i,N}^{(l)}\left(\mathcal{G}_{\text{body}(\rho)\wedge\text{head}(\rho)}\right)$ has strictly lower than $\varepsilon'$-uncertainty. We conclude that pattern counts obtained from Algorithm 1 have a strictly lower uncertainty than pattern counts obtained from random walks for the same $N(\varepsilon')$. Hence, by the result of stage 2, we can guarantee $\varepsilon$-uncertainty for the rule-set utility using Algorithm 1 with $N = N(\varepsilon)$ given by equation equation 6. The scaling law is, therefore, worst case,

$$N(\varepsilon) \sim \mathcal{O}\left(\frac{MD}{\varepsilon^2}\right),$$

and Algorithm 1 does strictly better than random walks. This concludes stage 3 and concludes the proof.

$\square$

**Remark 2.** *Theorem 3 is a worst-case upper bound. For instance, for homogeneous data, the upper bound scaling is $N \propto \mathcal{O}\left(\frac{MD}{|V|\varepsilon^2}\right)$. In our experiments (Section 7), we find that setting $N = \frac{MD}{|V|\varepsilon^2}$ performs well when all nodes in the data have roughly the same binary degree.*

### A.4 Justification of Remark 2

In the above proof of Theorem 3 we considered the worst-case scenario, where we required $\varepsilon$-uncertainty of top-$M$ pattern fragments found locally around *each* node $v_i$ (c.f. stage 1 of the proof). In reality, in many datasets, rule fragments that appear in the $D$-neighbourbood of one node, will also appear within the $D$-neighbourhoods of many other nodes in the data graph $\overline{\mathcal{G}}_\mathcal{D}$. The limiting case is the case of homogeneous data, where the pattern probabilities in the $D$-neighbourhood of every node in $\mathcal{D}_G$ are the same. In this scenario, it is the sum of pattern counts from running random walks from *all* nodes that needs to be connected to the notion of $\varepsilon$-uncertainty. For a dataset with $|V|$ nodes, this means that the number of random walks required to run from each individual node is smaller by a factor of $|V|$, i.e. $N(\varepsilon) \sim \mathcal{O}(MD/\varepsilon^2|V|)$, as stated in Remark 2.

## B Utility Example

**Example 1.** *Let us consider recommender systems, where the goal is to predict whether a user will like an item based on user and item characteristics and previous user ratings for other items. Let us assume the following background knowledge in first-order logic:*

$$\rho_1 : \text{FRIENDS}(U_1, U_2) \wedge \text{LIKES}(U_1, I) \rightarrow \text{LIKES}(U_2, I),$$

*which states that if two users $U_1$ and $U_2$ are friends and one user liked an item $I$, then the other user will also like the same item.*

***Symmetry factor calculation.*** *Assume that the training data $\mathcal{D}$ includes the facts:* LIKES(alice, starwars), FRIENDS(alice, bob), LIKES(bob, starwars). *Then, there are two ground patterns of* LIKES$(U_1, I) \wedge$ FRIENDS$(U_1, U_2)$, *and one ground pattern of* LIKES$(U_1, I) \wedge$ FRIENDS$(U_1, U_2) \wedge$ LIKES$(U_2, I)$ *in $\mathcal{D}$. Hence, for rule $\rho_1$, we obtain $\mathsf{P}(\rho_1) = \frac{1}{2}$ despite that the rule correctly predicts that* alice *likes* starwars *given that* bob *likes* starwars *as well as vice versa. However, this rule has a symmetry factor of 2, (as illustrated by Figure 1) and once the precision is corrected, we get $\mathsf{P}(\rho_1) \cdot \mathsf{S}(\rho_1) = 1$, as expected since the rule is always satisfied.*

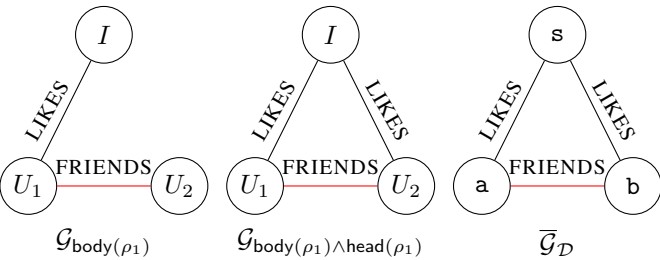

Figure 1: Datagraph $\overline{\mathcal{G}}_{\mathcal{D}}$ for a dataset $\mathcal{D} = \{$ LIKES(alice, starwars), FRIENDS(alice, bob), LIKES(bob, starwars)$\}$. Constants alice, bob, and starwars are abbreviated as a, b, and s. For this data, rule $\rho_1$ has a single grounding. However, the number of isomorphisms of $\mathcal{G}_{\mathsf{body}(\rho_1)}$ into $\mathcal{G}_{\mathsf{body}(\rho_1) \wedge \mathsf{head}(\rho_1)}$ is 2, hence $\mathsf{S}(\rho_1) = 2$.

***Bayes factor calculation.*** *Consider, in addition to rule $\rho_1$, rule $\rho_2$:*

$$\rho_2 : \text{LIKES}(U_1, I) \wedge \text{FRIENDS}(U_1, U_2) \rightarrow \text{DISLIKES}(U_2, I).$$

*Assume also that the facts $\mathcal{D}$ abide by the following statistics: (i) if a user $U_1$ likes an item $I$, then a friend of theirs $U_2$ likes $I$ with probability 50% and dislikes $I$ with probability 50%, and (ii) the number of DISLIKES-facts is ten times larger than the number of LIKES-facts. From assumption (i) and Definition 1, it follows that $\mathsf{P}(\rho_1) = \mathsf{P}(\rho_2) = \frac{1}{2}$ since for each grounding of the body* LIKES$(u_1, i) \wedge$ FRIENDS$(u_1, u_2)$ *in $\mathcal{D}$, there is either a fact* LIKES$(u_2, i)$ *or a fact* DISLIKES$(u_2, i)$ *in $\mathcal{D}$ and each fact has a probability of 50%. If there were no correlations in the facts (i.e. assumption (i) does not hold), then we would expect, from assumption (ii), that the head is ten times more likely to be DISLIKES than LIKES. Therefore, the result $\mathsf{P}(\rho_1) = \mathsf{P}(\rho_2) = \frac{1}{2}$ is misleading, since $\rho_1$ is correct over five times more often than random chance (1/2 vs 1/11) and $\rho_2$ is correct less often than random chance (1/2 vs 10/11).*

*We now compute the Bayesian priors for these rules. In both cases, the head predicate contains a user and an item term, thus, $\mathcal{A} = \{\text{LIKES}(U, I), \text{DISLIKES}(U, I)\}$ for both rules (Definition 3). The Bayesian priors are thus:*

$$\mathsf{B}(\rho_1) = \frac{|\overline{\mathcal{G}}_{\text{LIKES}(U,I)}|}{|\overline{\mathcal{G}}_{\text{LIKES}(U,I)}| + |\overline{\mathcal{G}}_{\text{DISLIKES}(U,I)}|} = \frac{1}{10+1}, \quad \mathsf{B}(\rho_2) = \frac{|\overline{\mathcal{G}}_{\text{DISLIKES}(U,I)}|}{|\overline{\mathcal{G}}_{\text{LIKES}(U,I)}| + |\overline{\mathcal{G}}_{\text{DISLIKES}(U,I)}|} = \frac{10}{10+1}.$$

*Notice that these are exactly the probabilities of $\rho_1$ and $\rho_2$ being true if the facts in $\mathcal{D}$ were uncorrelated. Precision of the rules, corrected for this uniform prior, would then be $\frac{\mathsf{P}(\rho_1)}{\mathsf{B}(\rho_1)} = \frac{11}{2}$ and $\frac{\mathsf{P}(\rho_2)}{\mathsf{B}(\rho_2)} = \frac{11}{20}$. Since the first ratio is larger than one, rule $\rho_1$ successfully predicts a correlation. In contrast, for rule $\rho_2$, since this ratio is smaller than one, the rule makes a prediction that is worse than a random guess. Hence, by this metric, rule $\rho_1$ is correctly identified as more useful than $\rho_2$.*

## C    PATTERN MINING EXAMPLE

**Example 2.** *We illustrate how Algorithm 1 mines patterns from the graph $\overline{\mathcal{G}}_{\mathcal{D}}$ shown in Figure 2. We follow a recursive call from node $v_0$ with parameters $N = 4$ and $D = 2$. To ease the presentation, we denote ground patterns as sets of edges.*

*Since $e_0$ is the only unary edge of $v_0$, Algorithm 1 stores the pattern $\{e_0\}$ in $\overline{\mathcal{G}}_{\mathsf{global}}$. Algorithm 1 then finds two binary edges $e_1$ and $e_2$ and, since $2 \leq N$, it considers both edges in turn. Algorithm 1*

*then grafts these two edges onto the previous pattern which by default contains the empty pattern $\emptyset$. In particular, for $e_1$, we graft $\emptyset \circ e_1 = \{e_1\}$ and $\{e_0\} \circ e_1 = \{e_0, e_1\}$. The resulting patterns along with $\mathcal{E}_{\text{previous}} = \{e_1\}$ are passed as*

$$\overline{\mathcal{G}}_{\text{final}} = \left\{ \begin{array}{c} \{e_1\} \\ \{e_0, e_1\} \end{array} \right\}$$

*to the next recursive call and are stored in $\overline{\mathcal{G}}_{\text{global}}$. Algorithm 1 then proceeds analogously along $e_2$. After visiting node $v_2$, $\overline{\mathcal{G}}_{\text{global}}$ has as follows:*

$$\overline{\mathcal{G}}_{\text{global}} = \left\{ \begin{array}{c} \{e_0\} \\ \{e_1\} \\ \{e_2\} \\ \{e_0, e_1\} \\ \{e_0, e_2\} \end{array} \right\}.$$

*Both new recursions are started with updated $n = 4/2$ and $d = 1$. At $v_1$, since $n = 2$ and since there are two, previously unvisited, edges $e_4$ and $e_5$, the algorithm continues the recursion along $e_4$ and $e_5$ with updated $n = 2/2$ and $d = 2$. In contrast, at $v_2$, since $n = 2$ but there being three, previously unvisited, edges $e_7$, $e_8$, and $e_9$, the algorithm will choose at random two out of the three edges and continue the recursion with $n = 2/2$. Since, $d = 2$ in the next recursive calls, the algorithm terminates. Notice that if $N$ was set to six, then Algorithm 1 would have found all ground patterns $\overline{\mathcal{G}}_{\mathcal{D}}$.*

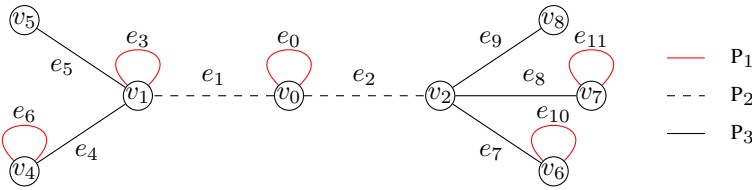

Figure 2: Graph $\overline{\mathcal{G}}_{\mathcal{D}}$ from Example 2. The graph contains three types of labelled edges: red unary edges $P_1$, dashed black binary edges $P_2$, and solid black binary edges $P_3$.

# D EXPERIMENTAL DETAILS

## D.1 LEARNING MARKOV LOGIC NETWORKS

**Datasets.** We consider two benchmark datasets for learning MLNs (Richardson & Domingos, 2006): the IMDB dataset, which describes relationships among movies, actors and directors, and the WEBKB dataset, consisting of web pages and hyperlinks collected from four computer science departments. Each dataset has five splits.

Table 3: Data statistics of benchmark MLN datasets.

| Dataset | Ground Atoms | Relations |
|---------|--------------|-----------|
| IMDB    | 980          | 10        |
| WEBKB   | 1,550        | 6         |

**Problem.** The task is to infer truth values for missing data based on partial observations. For example, in the IMDB dataset, we might not have complete information about which actors starred in certain movies. In this case, our objective would be to estimate, for each actor, the probability that they appeared in a particular film by performing inference over the observed data with a learnt MLN model. The missing data covers all predicates in the database, such as STARRINGIN(movie, person), ACTOR(person), GENRE(movie) etc. This problem can thus be framed as predicting missing links in a (hyper)graph.

**Experimental setup.** For all SPECTRUM experiments, we set $N = \frac{MD}{|V|\varepsilon^2}$ (see Remark 2), $M = 30$, $\varepsilon = 0.1$, and $D = 3$, running them on a 12-core i7-10750H CPU.

**Results.** We compared against LSM (Kok & Domingos, 2010), BOOSTR (Khot et al., 2015), and PRISM (Feldstein et al., 2023b), using the parameters as suggested by the respective authors. Since PRISM only mines motifs, we used LSM for the remaining steps of the pipeline, in line with Feldstein et al. (2023b). We used ALCHEMY (Kok et al., 2005) – an implementation of an MLN framework – to calculate the averaged conditional loglikelihood on each entity (ground atom) in the test split. We perform leave-one-out cross-validation and report the average balanced accuracy (ACC) and runtimes in Table 4 . SPECTRUM improves on all fronts: the runtime is $< 1\%$ compared with the most accurate prior art, while also improving accuracy by over 16% on both datasets.

Table 4: Balanced accuracy (ACC) and runtime comparisons of PRISM, LSM, BOOSTR, and SPECTRUM on MLN experiments.

|  | Algorithm | ACC | RUNTIME (s) |
|---|---|---|---|
| IMDB | LSM | $0.55 \pm 0.01$ | $430 \pm 20$ |
|  | BOOSTR | $0.50 \pm 0.01$ | $165.7 \pm 129$ |
|  | PRISM | $0.58 \pm 0.01$ | $320 \pm 40$ |
|  | SPECTRUM | $\mathbf{0.74 \pm 0.02}$ | $\mathbf{0.8 \pm 0.05}$ |
| WEBKB | LSM | $0.65 \pm 0.005$ | $220 \pm 10$ |
|  | BOOSTR | $0.12 \pm 0.09$ | $9.3 \pm 0.4$ |
|  | PRISM | $0.65 \pm 0.005$ | $102 \pm 5$ |
|  | SPECTRUM | $\mathbf{0.81 \pm 0.01}$ | $\mathbf{0.5 \pm 0.02}$ |

### D.2 SCALABLE LEARNING OF LOGICAL MODELS

**Datasets.**

1. Citeseer: This dataset consists of research papers, categories the papers fall under, and links between papers. Citeseer has the relations HASCAT($P, C$) (describing whether a paper $P$ is of a specific category $C$) and LINK($P_1, P_2$) (describing whether two papers are linked). The dataset has six paper categories.

2. Cora: Cora is also a citation network of papers, equivalent to Citeseer except having seven categories.

3. CAD: The collective activity detection dataset (CAD) contains relations about people and the actions (waiting, queuing, walking, talking, etc.) they perform in a sequence of frames. FRAME($B, F$) states whether a box $B$ (drawn around an actor in a frame) is in a specific frame $F$; FLABEL($F, A$) states whether most actors in a frame perform action $A$; DOING($B, A$) states whether an actor in a box $B$ performs action $A$; CLOSE($B_1, B_2$) states whether two boxes in a frame are close to each other; SAME($B_1, B_2$) states whether two bounding boxes across different frames depict the same actor.

4. Yelp: The Yelp 2020 dataset contains user ratings on local businesses, information about business categories and friendships between users. We used the pre-processing script proposed by Kouki et al. (2015) to create a dataset consisting of SIMILARITEMS, SIMILARUSERS, FRIENDS, AVERAGEITEMRATING, AVERAGEUSERRATING, and RATING, describing relations between users and items.

**Experimental setup.** For all experiments, we set $N = \frac{MD}{|V|\varepsilon^2}$, $M = 30$, $\varepsilon = 0.1$, and $D = 3$, running them on a 12-core i7-10750H CPU, except for the CAD experiment where we sat $M = 60$ due to the large number of different predicates.

**Recovering hand-crafted PSL rules.**

Table 5: Data statistics of training sets for PSL scalability experiments.

| Dataset | Ground Atoms | Relations |
|---------|--------------|-----------|
| Citeseer | $6,800$ | 2 |
| Cora | $6,900$ | 2 |
| CAD | $250,000$ | 19 |
| Yelp | $2,200,000$ | 5 |

*Citeseer and Cora*: For each category, we introduce the following relation HASCATX(P), where X refers to the category. This allows us to find different rules for different categories. For each dataset, SPECTRUM finds the following rules:

$$\text{HASCAT}(\text{P}_1, c) \wedge \text{LINK}(\text{P}_1, \text{P}_2) \rightarrow \text{HASCAT}(\text{P}_2, c),$$

for every $c \in \{1, \ldots, 6\}$ for Citesser and $c \in \{1, \ldots, 7\}$ for Cora. Importantly, the engineering back to constants for the categories, e.g. HASCAT1(P) to HASCAT(P$_1$, 1), is implemented in SPECTRUM, and can be applied for any categorical value. This automatic translation is very useful in classification tasks.

*CAD*: We perform pre-processing akin to Citeseer and Cora and introduce DOINGX(B) relations for each action X. SPECTRUM then finds the same 21 rules as hand-engineered by London et al. (2013). Namely, we get the following three rules for every action $a \in$ {crossing, waiting, queuing, walking, talking, dancing, jogging}:

$$\text{FRAME}(\text{B}, \text{F}) \wedge \text{FLABEL}(\text{F}, a) \rightarrow \text{DOING}(\text{B}, a)$$
$$\text{DOING}(\text{B}_1, a) \wedge \text{CLOSE}(\text{B}_1, \text{B}_2) \rightarrow \text{DOING}(\text{B}_2, a)$$
$$\text{DOING}(\text{B}_1, a) \wedge \text{SAME}(\text{B}_1, \text{B}_2) \rightarrow \text{DOING}(\text{B}_2, a)$$

*Yelp*: We split the RATING relations into RATINGHIGH and RATINGLOW to see whether SPECTRUM identifies differences between how high and low ratings are connected. We find the following six rules:

$$\text{RATINGX}(\text{U}_1, \text{I}) \wedge \text{SIMILARUSERS}(\text{U}_1, \text{U}_2) \rightarrow \text{RATINGX}(\text{U}_2, \text{I})$$
$$\text{RATINGX}(\text{U}, \text{I}_1) \wedge \text{SIMILARITEMS}(\text{I}_1, \text{U}_2) \rightarrow \text{RATINGX}(\text{U}, \text{I}_2)$$
$$\text{RATINGX}(\text{U}_1, \text{I}) \wedge \text{FRIENDS}(\text{U}_1, \text{U}_2) \rightarrow \text{RATINGX}(\text{U}_2, \text{I}),$$

where $X \in \{\text{high}, \text{low}\}$. The only rules hand-engineered by Kouki et al. (2015) that SPECTRUM cannot find are

$$\text{AVERAGEITEMRATING}(\text{I}) \leftrightarrow \text{RATING}(\text{U}, \text{I})$$
$$\text{AVERAGEUSERRATING}(\text{U}) \leftrightarrow \text{RATING}(\text{U}, \text{I}),$$

since these rules are not term-constrained.

### D.3 KNOWLEDGE GRAPH COMPLETION

**Experimental setup.** For all SPECTRUM experiments, we set $N = \frac{MD}{|V|\varepsilon^2}$, $M = 20 \times$ [number of relations], $\varepsilon = 0.01$, and $D = 3$, running them on a 12-core i7-10750H CPU. All NCRL and RNNLogic experiments were run on a V100 GPU with 30Gb of memory. For these models, we used the same hyperparameters as suggested by the authors in the original papers.

**Dataset statistics for knowledge graph completion**.

Table 6: Data statistics of training sets for benchmark knowledge graph datasets.

| Dataset | Ground Atoms | Relations |
|---------|--------------|-----------|
| Family | 5,868 | 12 |
| UMLS | 1,302 | 46 |
| Kinship | 2,350 | 25 |
| WN18RR | 18,600 | 11 |
| FB15K-237 | 68,028 | 237 |

