# OpenReview forum: "Efficiently Learning Probabilistic Logical Models by Cheaply Ranking Mined Rules"
_ICLR.cc/2025/Conference — Submitted to ICLR 2025_

### Official Review · Reviewer_gjdo · 2024-10-28

**Soundness:** 2
**Presentation:** 3
**Contribution:** 3
**Rating:** 6
**Confidence:** 3

**Summary:**

This paper aims at rule mining in the neuro-symbolic (NS)  domain. Rule mining is critical in NS methods which typically requires domain-specific rules. To address this, the paper defines several criteria to measure the utility of various rules and develop an algorithm for fast rule mining. They investigate the performance in various datasets: for MLN datasets, the method successfully mined most rules; for graph completion datasets, the method effectively outperforms RNNLogic and NCRL, two typical NS methods.

**Strengths:**

- This paper is well-written and well-structured. The reader can quickly understand the main idea of the paper.
- The proposed method operates on CPU while still getting comparable results against other methods, demonstrating strong application value in the industry.

**Weaknesses:**

I have several questions:

-  The "precision" measure seems only to consider horn-clause, however, in MLNs, we also consider more general rules. In other words, 'a->b' also means 'not b-> not a'. Considering 'not b -> not a' is also important in some scenarios. How do you address this problem?
- Throughout the paper, I do not find a reference of AMIE [1], a fast and effective method of rule mining also using heuristic measurements. How do you compare with AMIE?
- How do you leverage the mined rules in the link completion tasks? Do you use some linear models? I did not find the details of the task in sec 7.2.

[1] AMIE3: https://luisgalarraga.de/docs/amie3.pdf

**Questions:**

Please see the questions above.

---

### Official Review · Reviewer_Zbym · 2024-10-28

**Soundness:** 4
**Presentation:** 3
**Contribution:** 3
**Rating:** 6
**Confidence:** 3

**Summary:**

This paper introduces a framework called SPECTRUM, which focuses on efficiently learning probabilistic logical models to address scalability challenges in neurosymbolic AI. SPECTRUM incorporates a low-cost utility measure for assessing the predictive performance of logical models. Additionally, it employs a linear-time pattern mining algorithm to identify common structures within data and uses a fast sorting algorithm to optimize rule selection. Experimental results demonstrate that SPECTRUM can quickly construct efficient logical models across various tasks, significantly reducing computation time while surpassing existing methods in terms of interpretability and accuracy.

**Strengths:**

1. The paper has a well-defined motivation for tackling scalability issues in probabilistic logic models, with concise and clear explanations.

2.  The authors provide theoretical guarantees on the computational cost required for a certain error bound on the utility estimates.

3.  Experimental results show that SPECTRUM significantly outperforms previous methods in both efficiency and predictive accuracy, highlighting its practical advantages

**Weaknesses:**

1. The SPECTRUM framework itself does not learn probabilistic rules, and the emphasis on "probabilistic" throughout, including in the title, seems somewhat overstated.

2. The authors' introduction of Inductive Logic Programming and Differentiable Rule Learning in the related work section appears insufficient. Some advanced differentiable rule learning algorithms have less reliance on templates and offer better scalability compared to traditional methods[1]; a comparison with these should be included.

[1] Glanois, Claire, et al. "Neuro-symbolic hierarchical rule induction." International Conference on Machine Learning. PMLR, 2022.

**Questions:**

Why use a quadratic-time greedy optimisation algorithm after getting M candidate rules? Most of the time, M is in the range of tens to hundreds. What will happen if a more precise optimisation algorithm is used to select the best subset? Will it really become a bottleneck for scalability (because your scalability focuses on the size of the data rather than M)?

---

### Official Review · Reviewer_jm86 · 2024-10-29

**Soundness:** 3
**Presentation:** 2
**Contribution:** 3
**Rating:** 6
**Confidence:** 2

**Summary:**

This paper discussed a method to evaluate the quality of a symbolic first-order rule. In addition, the paper also proposed a new method for mining rules from data. The measurement is cheap, the rule learning is in linear-time complexity, and the finding best utility algorithm is in quadratic-time complexity. The experiment results indicate that the model can be used in the ILP domain for learning rules and the model can perform knowledge graph completion for larger knowledge graphs such as the FB15K-237 dataset.

**Strengths:**

1. The structure of the paper is easy to understand;
2. The authors proposed well-designed metrics to evaluate the symbolic rule.
3. The proposed ILP algorithms are analyzed in terms of complexity and completeness. In addition, the experiments are conducted explicitly.

**Weaknesses:**

1. Some definitions are not clear to present. Please see the question 1.
2. For the scalability of the proposed ILP model, there are no results to indicate SPECTRUM can extract rules from very large knowledge graphs such as UMLS, Kinship, FB15K-237, etc. However, the  NeuralLP (Yang et al., 2017) and DRUM (Sadeghian et al., 2019) did learn symbolic rules from these large datasets.

**Questions:**

1. For the definition of isomorphic, the notation $v_i$ represents a node in a graph in line 99. However, $v_i$ represents an edge in line 106. The ‘instantiations of a rule’ is not defined in line 163. In addition, the ‘symmetries’ in line 144 is not defined.
2. When calculating the precision of a rule for example in line 983, what is the difference between ground patterns and substitutions which has a standard definition in the ILP community?
3. There are some missing references and comparisons when defining metrics such as precision and recall for a rule. For some work [1,2,3], they also discussed precision, recall, rule length, etc. Can you evaluate some differences and similarities of your definition above the precision, recall, and complexity factor with them?

Reference:

[1] Tim Rocktäschel, Sebastian Riedel: End-to-end Differentiable Proving. NIPS 2017: 3788-3800

[2] Kun Gao, Katsumi Inoue, Yongzhi Cao, Hanpin Wang: A differentiable first-order rule learner for inductive logic programming. Artif. Intell. 331: 104108 (2024)

[3] Litao Qiao, Weijia Wang, Bill Lin: Learning Accurate and Interpretable Decision Rule Sets from Neural Networks. AAAI 2021: 4303-4311

---

### Official Review · Reviewer_GgSo · 2024-11-04

**Soundness:** 3
**Presentation:** 3
**Contribution:** 2
**Rating:** 3
**Confidence:** 4

**Summary:**

This paper deals with the task of efficiently learning logical rules from large relational databases. This can be seen as structure learning for probabilistic (and neural) logical models and hence efficient rule learning is also an important problem in statistical relational and neuro symbolic AI. The authors approach this problem by proposing the SPECTRUM, an efficient rule learning algorithm based on a 2-phase process. The first phase is pattern mining; it involves finding frequent relational patterns from the database. The second phase is pattern optimization; it involves (greedily) finds the top-M rules according to an efficiently computable rule utility score. The authors prove completeness and approximate optimality of the pattern mining phase and derive a principled yet efficiently computable utility function to score theories. Once the rules are learned, they can be used to instantiate a statistical relational or neurosymbolic model. The authors consider the former case and specifically instantiate probabilistic soft logic (PSL) models. They compare SPECTRUM against NCRL (Cheng et al., 2023) and RNNlogic (Qu et al., 2021) on 5 knowledge graph completion databases. The results show that SPECTRUM outperforms the baselines especially on larger databases. The authors also compare SPECTRUM against MLN learning methods including LSM (Kok & Domingos, 2010), BOOSTR (Khot et al., 2015), and PRISM (Feldstein et al., 2023) on IMDB and WebKB databases, showing that SPECTRUM outperforms the baselines.

**Strengths:**

- The problem of scalable rule learning is highly important
- SPECTRUM does appear to be more efficient and scalable as compared to baselines.
- The authors formally prove a number of properties of SPECTRUM including completeness and approximate optimality of the pattern mining phase

**Weaknesses:**

The paper is missing key related work and baselines:

1.	Poole et al., (2014) diagnose the scalability problems of MLNs.
2.	Gradient boosted Relational Dependency Networks (Natarajan et al., 2012) perform joint structure and parameter learning and have been shown to outperform MLNs
3.	B-RLR (Ramanan et al., 2021) learns Relational Logistic Regression models (Kazemi et al., 2014) using Functional Gradient boosting
4.	NNRPT (Kaur et al., 2020) uses relational random walks to instantiate neural networks with rule-based parameter tying.
5.	alpha-ILP (Shindo et al., 2023) uses k-beam search with a differentiable reasoner to learn rule weights.


Poole, D., Buchman, D., Kazemi, S. M., Kersting, K., & Natarajan, S. (2014). Population size extrapolation in relational probabilistic modelling. In Scalable Uncertainty Management: 8th International Conference, SUM 2014

Kaur, N., Kunapuli, G., Joshi, S., Kersting, K., & Natarajan, S. (2020). Neural networks for relational data. ILP.

Natarajan, S., Khot, T., Kersting, K., Gutmann, B., & Shavlik, J. (2012). Gradient-based boosting for statistical relational learning: The relational dependency network case. Machine Learning, 86

Shindo, H., Pfanschilling, V., Dhami, D. S., & Kersting, K. (2023). α ILP: thinking visual scenes as differentiable logic programs. Machine Learning, 112(5)

Ramanan, N., Kunapuli, G., Khot, T., Fatemi, B., Kazemi, S.M., Poole, D., Kersting, K. and Natarajan, S. (2021). Structure learning for relational logistic regression: an ensemble approach. Data Mining and Knowledge Discovery, 35

Kazemi, S.M., Buchman, D., Kersting, K., Natarajan, S. and Poole, D., (2014). Relational logistic regression. KR

**Questions:**

How does the proposed method (SPECTRUM) compare to the prior work listed above?

---

### Author Response · Authors · 2024-12-03
**Summary of review discussions**

We would like to summarise the review discussions for the benefit of the AC. We also like to thank the reviewers again for their comments and suggestions as we believe these reviews helped improve the paper from the original submission.

**Reviewer GgSo**: The only remark reviewer GgSo made was on missing citations and baselines to prove SPECTRUM's scalability. We summarize our analysis of the baselines they asked us to compare against:

* Natarajan, S (2012) RDNs: We ran RDN's rule learning algorithm on Family, UMLS, WN-18RR (see Table 2) using the codebase at https://github.com/hayesall/rfgb. **The RDN's rule learning algorithm needs $>$15 minutes to learn rules for a single predicate, while SPECTRUM takes at most 36 seconds to learn rules and their weights for all predicates in these benchmarks, see Table 2.**

* B-RLR (Ramanan et al., 2021 and Felix Weitkämper, 2024). The B-RLR implementation at https://github.com/nandhiniramanan5/RLR_Boost cannot be executed successfully, as also reported in Section 5 by (Felix Weitkämper, 2024). Similarly, the B-RLR implementation at https://github.com/srlearn/srlearn throws exceptions as reported in our earlier replies. **Regarding the B-RLR implementation at (Felix Weitkämper,2024) we contacted the authors, who replied that B-RLR does not find meaningful rules for the link prediction datasets from Table 2.**

* $\alpha$ILP (Shindo et al., 2023): $\alpha$ILP is a rule learning technique for a specific visual task, applied to scenarios including $\sim1000$ facts. In contrast, SPECTRUM runs on benchmarks of millions of facts (see benchmarks in Section 7.1). Further, **the inputs to $\alpha$ILP are *images*, and the code base is implemented for specific visual tasks/datasets, making it unclear how to use it for the benchmarks in Section 7 and Appendix D. We asked reviewer GgSo for their suggestions, but they did not provide any.**

As we pointed out in our rebuttal, the techniques suggested by reviewer GgSo (i) were originally tested by their authors on smaller datasets than the ones in our paper and (ii) are even significantly slower than the baselines in Section 7.2. Based on the above, if reviewer GgSo believes that a comparison against his suggestions is needed to prove scalability, then this has been shown beyond any doubt. Hence, it is unclear why in this case the grade remains a 3/10.


**Reviewer jm86**: The reviewer found two weaknesses:

1. Unclear definitions: We have cleaned up the definitions mentioned by the reviewer (seemingly to the reviewer's satisfaction).

2. Missing experiments: The experiments the reviewer claimed were missing were already in Section 7.2 and we have pointed this out in the rebuttal.

Based on (i) having cleaned the definitions, (ii) the experiments being present in the paper, it is unclear why the grade has not changed from the initial 6/10, since no additional weaknesses were added.

**Reviewer Zbym**: The reviewer found two weaknesses:

1. They claimed that our work does not learn probabilistic models: We provided a list of arguments explaining why this statement is false (seemingly to the reviewer's satisfaction).

2. Missing comparison (Glanois 2022): We performed a detailed analysis (theoretical and experimental) and found that the work by (Glanois 2022) does not scale even to the smallest datasets we tested, making a comparison impossible.

Since (i) the above issues seem to be answered, (ii) no further weaknesses have been mentioned, and (iii) the individual grading is very positive (Soundness: 4 excellent, Presentation: 3 good, Contribution: 3 good),  it is unclear to us why the grading has not improved from the 6/10.

**Reviewer gjdo**: The reviewer identified three weaknesses:

1. SPECTRUM is limited to Horn clauses: As we pointed out, this assumption is made by the vast majority of  structure learning techniques, and, in particular, by the latest state-of-the-art frameworks we compare against. In addition, we presented a solution to engineer around this limitation.

2. A comparison to AMIE3 is missing: We provided an in-depth (theoretical and experimental) analysis of AMIE3.

3. Confusion about how inference was performed in experiments: We pointed the reviewer to the paragraph in the original submission explaining this.

The reviewer increased their score from a 5/10 to a 6/10 for comparing against AMIE3 but provided no comments on the other points. It is unclear to us what weaknesses remain from their perspective. Seemingly, there are issues in soundness (scored with a 2) but the reviewer offered no explanations. We believe our six pages with proofs to be correct and despite us requesting further clarification, scoring soundness with 2 is unanswered.

---

### Meta-Review · Area_Chair_jPGP · 2024-12-24

**Metareview:**

This paper introduces a framework called SPECTRUM, a method to learn  probabilistic logical model from large relational databases. In terms of the proposed method itself, more than half of the reviewers share the appreciation of its technical contribution. Furthermore, the method first proposes a metric to evaluate the performance and then employs a linear-time pattern mining algorithm to identify common structures and uses a sorting algorithm to do the selection. The linear-time complexity has theoretical guarantee, which is a plus. However, firstly, this paper is not well contextualized; a few important references are not well discussed.  Second, the experiment results are not enough to convince readers. I would encourage the paper to, instead of spending more time on running experiments, put more efforts to carefully discuss about its baselines, and build a stronger case about its evaluation protocol. After that, to re-evaluate what experiments remain and what need to be added or ru-run.

**Additional Comments On Reviewer Discussion:**

Two main concerns. First, it is regarding references. This point has been mainly addressed during the rebuttal. The second concern, is whether the comparison can fully justify the proposed method's benefit. Reviewers have concern regarding the fairness of the comparison with baselines, and this concern still persists after the rebuttal.

---

### Decision · Program_Chairs · 2025-01-22

Reject